# Biotests in Cyanobacterial Toxicity Assessment—Efficient Enough or Not?

**DOI:** 10.3390/biology12050711

**Published:** 2023-05-12

**Authors:** Petar Davidović, Dajana Blagojević, Jussi Meriluoto, Jelica Simeunović, Zorica Svirčev

**Affiliations:** 1Department of Biology and Ecology, Faculty of Sciences, University of Novi Sad, Trg Dositeja Obradovića 2, 21000 Novi Sad, Serbia; petar.davidovic@dbe.uns.ac.rs (P.D.); dajana.blagojevic@dbe.uns.ac.rs (D.B.); 2Faculty of Science and Engineering, Biochemistry, Åbo Akademi, Tykistökatu 6 A, 20520 Turku, Finland; jussi.meriluoto@abo.fi

**Keywords:** cyanobacteria, cyanotoxins, toxicity assessment, biotests

## Abstract

**Simple Summary:**

Cyanobacterial toxins (cyanotoxins) pose a threat to human, animal, and environmental health. This review article provides an overview of the challenges associated with the detection and characterization of cyanotoxins using various biotests. The article discusses the use of alternative aquatic model organisms for the assays and the need for a multi-level approach to studying cyanotoxicity. The authors suggest that continued development and refinement of assays is necessary for their improved detection, characterization, and risk assessment and emphasize the need for a multi-level approach to studying cyanotoxicity.

**Abstract:**

Cyanobacteria are a diverse group of organisms known for producing highly potent cyanotoxins that pose a threat to human, animal, and environmental health. These toxins have varying chemical structures and toxicity mechanisms and several toxin classes can be present simultaneously, making it difficult to assess their toxic effects using physico-chemical methods, even when the producing organism and its abundance are identified. To address these challenges, alternative organisms among aquatic vertebrates and invertebrates are being explored as more assays evolve and diverge from the initially established and routinely used mouse bioassay. However, detecting cyanotoxins in complex environmental samples and characterizing their toxic modes of action remain major challenges. This review provides a systematic overview of the use of some of these alternative models and their responses to harmful cyanobacterial metabolites. It also assesses the general usefulness, sensitivity, and efficiency of these models in investigating the mechanisms of cyanotoxicity expressed at different levels of biological organization. From the reported findings, it is clear that cyanotoxin testing requires a multi-level approach. While studying changes at the whole-organism level is essential, as the complexities of whole organisms are still beyond the reach of in vitro methodologies, understanding cyanotoxicity at the molecular and biochemical levels is necessary for meaningful toxicity evaluations. Further research is needed to refine and optimize bioassays for cyanotoxicity testing, which includes developing standardized protocols and identifying novel model organisms for improved understanding of the mechanisms with fewer ethical concerns. In vitro models and computational modeling can complement vertebrate bioassays and reduce animal use, leading to better risk assessment and characterization of cyanotoxins.

## 1. Introduction

Toxin production by certain species of cyanobacteria has been recognized for more than 140 years, ever since the characteristic adverse effects in higher organisms were first identified [1]. Toxic cyanobacterial blooms in water can lead to acute poisonings in humans and animals from direct or indirect contact, as documented in case reports worldwide [2,3,4,5]. Cyanobacterial cells contain toxic metabolites that can be released in high amounts when the cells are lysed. Along with well-known toxins, additional toxic compounds with cytotoxic or cytostatic properties, and even tumor-promoting activity, have been discovered in cyanobacteria [6,7]. It is important, however, to note that although cyanobacterial species with the potential to produce toxins may be present in a certain environment, it does not necessarily make the environment toxic, since toxins are not continually produced in high enough concentrations to present a health risk [8]. In fact, cyanobacterial toxicity is highly variable and is, as such, difficult to predict, even when the producing organism and its abundance in the environment have been identified. Furthermore, several strains have been shown to be capable of turning the toxicity-related genes on or off, depending on the environmental conditions [9]. Some of these environmental factors, predominantly nutrient availability, temperature, and light intensity, could therefore be important in determining toxin production, although this is yet to be fully explored. For these reasons, the development of precise and reliable methods for the detection and quantification of cyanobacterial toxicity in different environments is of great importance.

Another problem is presented by the sheer number of potentially harmful products cyanobacteria are able to biosynthesize. It is possible to roughly classify these metabolites according to their toxic mode of action into hepatotoxins, neurotoxins, cytotoxins, dermatotoxins, and irritant toxins (lipopolysaccharides, etc.) [10,11]. Hepatotoxic cyanotoxins are the most frequently found and best-studied class of cyanobacterial toxins. This group consists of relatively large molecules (MW 800–1100), among which are cyclic heptapeptide microcystins and pentapeptide nodularins [12]. Both of these families of toxins share the same basic structure, which includes a unique amino acid side chain known as ADDA (3-amino-9-methoxy-2,6,8-trimethyl-10-phenyldeca-4,6-dienoic acid), which is considered responsible for the toxic properties of these molecules [13]. The cyclic heptapeptide structure of microcystins allows for great structural variability, the result of which is a family of 279 related toxic substances [14]. Production of microcystins (MCs) has been reported in several genera of freshwater cyanobacteria including *Microcystis*, *Nostoc*, *Anabaena*, *Anabaenopsis*, and *Planktothrix* [15], while nodularin (NODLN) was characterized only in *Nodularia spumigena* [16]. The modes of action of both microcystins and nodularins are based on the inhibition of serine-threonine protein phosphatases type 1 and type 2A (PP1 and PP2A) in eukaryotic cells, resulting in the disruption of the cytoskeletal structures in liver cells, loss of shape, and their subsequent destruction, causing intrahepatic hemorrhage [17,18,19]. Oxidative stress is also increased due to MC activity, which can potentially trigger apoptotic processes in the affected cells [20].

Cylindrospermopsins (CYNs) are guanidine alkaloid toxins mainly associated with *Cylindrospermopsis raciborskii* [21], although other producers have been identified, including the species *Umezakia natans* and *Aphanizomenon ovalisporum* as well as various strains belonging to the genera *Raphidopsis*, *Anabaena*, and *Lyngbya*. While they are often classified as hepatotoxins, as the liver is their main target, these toxins affect multiple organ systems and differ from the previously described cyclic peptide group in the mechanism of action [22]. Research has shown that CYNs are potent inhibitors of protein synthesis, although this is not the only cause behind the toxicity of CYN, since other toxic effects related to CYN include genotoxicity via DNA fragmentation [23] and the depletion of reduced glutathione (GSH), even in low concentrations of CYN (0.8–1.6 µM) that are otherwise considered non-toxic, possibly through the inhibition of GSH synthesis [24]. Protein synthesis inhibition is an early indicator of the cellular response to CYN [25], as this effect is observed after exposure to subtoxic concentrations (0.5–5 µM) and in the early phase of toxicity caused by higher concentrations of the toxin.

The known neurotoxins produced by cyanobacteria include anatoxin-a, homoanatoxin-a, guanitoxin (formerly anatoxin-a(S) [26]), and saxitoxins. Anatoxins (ANTX) are water-soluble alkaloids involved in multiple incidents of human and animal poisonings, some of which have been fatal [4,27,28]. The production of anatoxin-a was first determined in a Canadian strain of *Anabaena flos-aquae* [29], and was named Very Fast Death Factor (VFDF) due to its reported deadliness to exposed livestock and wildlife. Since then, additional producers (*Anabaena circinalis*, *Aphanizomenon* sp., *Cylindrospermum* sp., and *Planktothrix* sp.) have been discovered [16]. Anatoxin-a and its homolog homoanatoxin-a are bicyclic secondary amines with molecular weights of 165 and 179 Da. Anatoxin-a acts by binding with high affinity to the neuronal nicotinic acetylcholine receptors, mimicking the action of acetylcholine and stimulating muscle cell contraction, which can result in respiratory muscle paralysis and death [30,31]. Guanitoxin (GTXN), found primarily in *A. flos-aquae* and *A. lemmermannii*, is a potent inhibitor of acetylcholine-esterase (AChE) [32]. The chemical structure is different from that of anatoxin-a, as GTXN is a guanidine methyl phosphate ester with a molecular weight of 252 Da [30]. The original addition of “S” in the name of the toxin was made because of the characteristic hypersalivation reported in intoxicated laboratory mice. Saxitoxins (STX) are tricyclic guanidium alkaloids with more than 50 naturally occurring analogs [33]. In fresh waters, they are mainly associated with the cyanobacterium *Aphanizomenon flos-aquae* [34]; however, several other genera have been known to produce these toxins including *Anabaena circinalis*, *Anabaena lemmermanni*, *Aphanizomenon flos-aquae*, *Cylindrospermopsis raciborskii*, *Lyngbya wollei*, and *Planktothrix* sp. [16,35]. Saxitoxins act by blocking the voltage-gated sodium channels found in cell membranes, thus preventing membrane depolarization and inhibiting the proliferation of the action potential crucial for muscle contraction [36,37].

Aquatic animal models have been proposed for the evaluation of the biological effects of cyanobacterial toxins, specifically the cyanotoxin exposure–effect relationship, as well as assessing human health risk and determining safe levels of exposure [38]. However, the efficiency of these and other proposed approaches in assessing cyanotoxin exposure–effect relationships and human health risks remains unclear. Given the increasing concerns regarding the risks associated with cyanobacterial toxins, there is a growing need for reliable and cost-effective alternatives to mammalian models for toxicity testing. Therefore, this review will examine the available evidence on the use of biotests in cyanobacterial toxicity assessment and assess their suitability as viable alternatives to mammalian models.

### Bioassays in Cyanobacterial Toxicity Testing

Models are used to help understand phenomena that are too complex or abstract for a direct approach. In environmental research, this could entail looking at the general forces at work in a specific ecosystem or predicting behaviors or interactions of certain elements of those systems. In such cases, it is often of use to look at specific living organisms responsive to even subtle changes in their environments. Although useful, models represent only close approximations and are never exact replicas of the phenomena they describe and their accuracy is largely dependent on the current state of scientific knowledge related to the subject in question [39]. For these reasons, new and improved models are constantly appearing as a result of scientific progress and discovery. The tests implementing living systems are called bioassays. In toxicology, bioassays are specific tests that consist of qualitative and quantitative measurements of toxic effects that are induced in model organisms after a certain period of exposure to a toxic agent [40]. They provide a convenient method for sample screening based on the biological response to toxic effects. The observed changes and the accumulated data are usually used for the estimation of the environmental impact of the substance in question, or to extrapolate the predicted human physiological response [41]. Today, various biological elements can be used in bioassays, from DNA and enzymes to cells, higher plants, invertebrates, and vertebrates. There are also many potential endpoints that can be investigated, from changes on a molecular level to cellular alterations, changes in specific physiological parameters, and even changes in motility and behavior [42].

Most studies concerned with the effects of cyanotoxins are conducted by observing changes occurring in a living organism after exposure to toxins in some form, either to a purified single toxin, crude extracts, or bloom biomass [43]. Studies on the occurrence, distribution, and frequency of toxic cyanobacteria were conducted in a number of countries during the 1980s using the mouse bioassay [44,45]. This assay, which earlier was a part of the routine procedure for monitoring cyanotoxins, has several and considerable downsides. Among the most prominent ones are the cost and maintenance issues; the assay also lacks sensitivity and specificity, especially in detecting cyanotoxins in low concentrations [46], and has often been starkly opposed on ethical grounds. Considerable efforts have therefore been made to discover and implement novel methods that would present an alternative to the mouse bioassay.

Some analytical techniques suitable for quantitative toxin determination became available in the late 1980s, and more were developed since then with increasing study of specific cyanotoxins [47]. Screening using various (quick and convenient) methods and trace analytical measurements of cyanotoxins complement each other. The first screening of a cyanobacterial sample for toxins is often conducted by either enzyme-linked immunosorbent assays (ELISA) or bioassays. Samples positive for cyanotoxins can then be analyzed for specific cyanotoxins using physico-chemical methods such as liquid chromatography–tandem mass spectrometry. The sensitivity of trace analytical methods is far superior to that of bioassays, especially when water samples are being analyzed, but the analytical methods require access to reference materials that are not commercially available for many cyanotoxins. While the sensitivity of bioassays is lower and it is not clear which compound(s) cause the toxic reaction in the test organisms/cells, the bioassays can basically cover a wide range of toxins, provided a suitable test organism/cell type is chosen. A comprehensive collection of theoretical overviews and standard operating procedures related to the sampling, screening, and trace analysis of cyanotoxins was recently published as a collaborative action of over 100 scientists working in the COST Action CYANOCOST [11]. Besides validation of the analytical robustness itself, the analyst should pay attention to receiving representative and non-tampered samples. Ideally, critical samples such as samples of potable water should be analyzed using two or more independent methods based on different chemical, biochemical, and toxicological principles.

In the literature available today, some of the most useful in vivo alternatives for the detection of cyanobacterial toxicity are presented via test procedures which involve invertebrate species such as the brine shrimp assay (*Artemia salina*), bioassay with daphnids (most commonly *Daphnia* sp. And *Ceriodaphnia* sp.), freshwater dipteran *Chironomus* sp., aquatic crustacean *Thamnocephalus platyurus*, as well as the embryo test with zebrafish (*Danio rerio*) and Japanese medaka fish (*Oryzias latipes*). Additionally, in vitro tests using cultured cells have been developed as an alternative to traditional whole-organism bioassays, allowing for the study of cyanotoxin effects on specific target tissues or organs in a controlled environment.

## 2. Bioassays with Vertebrate Animal Models

In the assessment of cyanobacterial toxicity, researchers often rely on vertebrate bioassays, which involve exposing living organisms to different levels of cyanotoxins and observing their physiological responses. Vertebrate bioassays offer several advantages over other methods of cyanotoxicity testing. They provide a more realistic and comprehensive representation of the effects of cyanotoxins on living organisms, taking into account factors such as metabolism, pharmacokinetics, and route of exposure [38]. Moreover, vertebrate bioassays might help to identify the specific mechanisms by which cyanotoxins cause toxicity, which can aid the development of effective treatment and prevention strategies [48]. Another advantage of vertebrate bioassays is their ability to simulate the effects of chronic exposure to cyanotoxins, which is often more relevant to real-world scenarios than acute exposure [49]. Despite their many benefits, vertebrate bioassays also have several limitations that must be considered. One significant challenge is the ethical concern associated with the use of live animals in research. Additionally, there is inter-species variability in the response to a toxin [9], which introduces a degree of uncertainty into the acquired data. Depending on whether a terrestrial or an aquatic model organism is selected, exposure routes can include oral ingestion or gavage, immersion in water containing purified cyanotoxins or crude extracts, or intraperitoneal injection [38,50]. The choice of exposure route can significantly impact the toxic potential of cyanotoxins and the severity of their effects. Studies have demonstrated notable differences in the effects caused by various exposure routes, emphasizing the importance of carefully selecting the appropriate model organism and exposure method for meaningful toxicity testing [51,52].

### 2.1. Mouse Bioassay

In this assay, adult mice are usually intraperitoneally (i.p.) injected or orally exposed to a sample and monitored for up to 24 h for toxicity-related symptoms. The observed symptoms and mortality are primarily used to determine the toxicity of bloom water samples in a qualitative manner, labeling a bloom as “toxic” or “non-toxic” [53,54,55], although the toxic symptoms and pathology observed in the exposed animals can indicate the class of the toxins [46,56]. Moreover, biological potency of the identified cyanotoxins can be established by calculating the LC50 values, and the assay can potentially be calibrated against a specific toxin such as microcystin-LR and, therefore, produce results in terms of microcystin-LR toxicity equivalents [46]. Mouse bioassay has been instrumental in determining the LC50 values of various types of cyanotoxins, including microcystins, cylindrospermopsin, nodularin, and saxitoxins (Table 1). The reported 24 h LC50 values for microcystin-LR, for example, range from 5 to 10.9 mg/kg [57,58,59,60] and 25 to 158 µg/kg [57,58,59,60,61,62] for oral and intraperitoneal administration, respectively. The liver and hepatocyte damage induced by both routes is similar, with hemorrhage and apoptosis being prominent [9,61,63,64]. Additionally, exposure to microcystin causes liver congestion with blood, increased liver/body weight ratios, and reduced serum glucose and total protein levels in the affected mice [58,60,65]. The LC50 values established for mice after nodularin exposure (intraperitoneal) range from 30 to 50 μg/kg [66,67,68]. Cylindrospermopsin, on the other hand, has been shown to cause cell death in mice, with a 24 h LC50 of approximately 2 mg/kg after intraperitoneal injection [21,68,69] and 0.2 mg/kg after 5–6 days of exposure [16]. Ingestion of CYN in mice can lead to liver damage, but it may also affect other organs, including the kidney, lungs, thymus, spleen, adrenal glands, and heart. This is especially true in cases where CYN-containing cell extracts are injected intraperitoneally, which can be more toxic and demonstrate a wider range of toxicity than pure CYN [68,70,71,72]. The toxicity of anatoxins appears to be largely dependent on the exposure route. Anatoxin-a and homoanatoxin-a have an LC50 range of approximately 200–250 µg/kg in mice (intraperitoneal), 730 µg/kg for dihydroanatoxin-a, and 20 µg/kg for guanitoxin [45,73]. However, when orally administered, the toxicity of dihydroanatoxin-a has been proven to be higher (feeding—8 mg/kg; gavage—2.5 mg/kg) than that of anatoxin-a (feeding—25 mg/kg; gavage—10 mg/kg) [72]. Finally, saxitoxins have LC50 values of 5–10 μg/kg (intraperitoneal) in mice [73]. As can be seen, much of the essential information on cyanobacterial toxicity and modes of action for different toxin groups available today were obtained using the mouse bioassay. Even today, this test remains a valuable tool for the initial screening of highly concentrated cyanobacterial samples of unknown toxicity. However, it is not appropriate for cyanotoxin quantification in water samples due to its lack of sensitivity and precision at low concentrations [46]. Additionally, the number of mice required for meaningful testing procedures would be impractical and is widely considered unethical. Therefore, an alternative method should be used for the sensitive quantification of cyanotoxins in water samples, particularly when testing for microcystins at concentrations around 1 μg/L, the value mentioned in the provisional drinking water guideline of the World Health Organization. Overall, the mouse bioassay can be useful in combination with other analytical methods for a comprehensive assessment of cyanobacterial toxicity.

#### Aquatic Vertebrate Animal Models

Fish species such as zebrafish (*Danio rerio*), fathead minnow (*Pimephales promelas*), rainbow trout (*Oncorhynchus mykiss*), Japanese medaka (*Oryzias latipes*), Nile tilapia (*Oreochromis niloticus*), and the common carp (*Cyprinus carpio*) are used as animal models in cyanobacterial toxicity studies, providing possible insights into the potential effects of toxins on human health. Smaller teleost fish species such as zebrafish and medaka are relatively easy to maintain and breed, allowing large numbers of individuals to be generated quickly for statistically significant sample sizes at low costs [74,75]. They also have a relatively short lifespan and develop rapidly, which simplifies the study of the effects of toxins on different developmental stages. Furthermore, some fish species, such as the zebrafish, show a high degree of genetic, anatomical, and physiological similarity with mammals [75]. This similarity allows researchers to investigate toxicological effects on fish organ systems and extrapolate the results to predict potential effects in humans with increased accuracy. Numerous studies have been conducted to investigate the effects of experimental intoxications in mammals and fish which have revealed significant liver damage, manifested as hemorrhages, apoptosis, cellular hypertrophy, and glycogen depletion, along with the occurrence of apoptotic cells and dissociation of liver sheets in response to microcystin exposure [63,76,77,78,79,80,81]. As is the case with mammalian organisms, fish liver is an ideal organ for studying cyanobacterial hepatotoxins, as it is the primary target of these toxins and also the principal detoxification organ, reflecting the organism’s overall response to xenobiotics. Moreover, liver function is tied to the regulation of reproduction in oviparous fish through the synthesis of hepatic vitellogenin (vtg) [80]. Therefore, fish liver proteome and transcriptome analysis can potentially provide insights into the reproductive toxicity of different cyanotoxins. In vivo studies indicate that microcystin exposure alters the morphology of fish liver cells [51,82,83,84]. However, though the morphological alterations found in fish livers following microcystin exposure resemble those produced in rat or mouse liver cells, there is evidence to suggest that fish may be less sensitive towards MC toxicity [85,86], possibly due to their long evolutionary history of adaptation to aquatic environments where they have been exposed to these toxins. Interestingly, toxicity studies have also suggested that embryos may be more susceptible to cyanobacterial toxins than juvenile and adult fish [87,88], although direct ambient environmental exposure of fish embryos to toxicants may be limited due to the highly resistant chorion of the embryo, which reduces microcystin penetration. Improved microinjection technologies have been developed [88] which are less aggressive to the embryos, resulting in very low mortality and providing a reliable means of exposure to toxicants for research purposes.

Fish, however, serve as more than just predictors of toxic agents’ direct impact on humans. Taking into account their critical role in both ecosystem function and human food provision, it is clear why an extensive amount of research has been conducted on cyanotoxin accumulation in fish muscle tissues, as well as histopathological changes in fish organs [38,89]. Fish, especially species that actively feed on phytoplankton, are directly exposed to possible hepatotoxins by ingesting contaminated organisms and/or passively via their epithelium (gills and skin) when hepatotoxins are dissolved in water [38]. Model organisms that incorporate both human and environmental health aspects are particularly valuable in the study of cyanotoxins given the connection between the two.

### 2.2. Danio Rerio Bioassay

The zebrafish (*Danio rerio*) is a well-recognized model organism in toxicological studies and is readily used in testing procedures, from those concerned with the effects on a molecular level to those tracking changes in the health and behavior of the organism itself [89,90,91,92]. The many advantages offered by this model organism include inexpensive maintenance, ease of manipulation due to small size, as well as easy breeding and high fecundity. One of the most useful characteristics, however, is related to the embryos of zebrafish, which develop ex utero, simplifying manipulation and toxin exposure [93]. Embryos are also transparent, which, alongside the fact that embryogenesis occurs relatively quickly, allows for the changes in morphological structures of their internal organs to be visualized easily using light microscopy. Another reason for the widespread use and acceptance of zebrafish embryos as an alternative to the experiments with mammalian models is the fact that embryos up to 96 hpf (hours postfertilization) do not fall under animal welfare regulation, as they are not considered equally aware of pain as adult individuals due to their still developing nervous system [94]. However, larval stages beyond this point can feed independently and are therefore protected. Moreover, zebrafish genome has been fully sequenced, revealing a high homology to the human genome [95], providing the means to connect early gene changes to those observed later in the toxic effects cascade at higher physiological levels such as the cellular, tissue, organ, and organism level. The recognition of these characteristics has focused the majority of studies concerned with the effects of exposure of zebrafish to toxic cyanobacterial metabolites on the early life stages of these organisms (Table 1). These traits make zebrafish a valuable animal model for monitoring toxin-induced changes throughout the complete developmental period of a vertebrate embryo.

#### 2.2.1. Developmental and Genotoxic Effects—*D. rerio*

Toxin mechanisms which target DNA structures or important cellular division and differentiation pathways can have a profound effect during the developmental phase of the exposed organism. This is especially true in aquatic environments, where the risk of chronic exposure of animals in the early stages of ontogenesis is arguably the highest. Many reports indicate that cyanobacterial toxins may induce significant changes in the embryo-larval development of aquatic organisms and propose the use of zebrafish as an adequate model organism for the investigation of such changes [96,97,98]. MC-LR in the concentration range of 0.5–50 µg/L has been shown to affect zebrafish survival in the larval stage of development. However, these effects did not occur during exposure, but only after larvae were transferred to clean water [90]. Research has also indicated that the crude cyanobacterial extracts could have much more severe effects on zebrafish than the effects of pure toxins [90], which was explained by the presence of as yet undefined toxic elements in cyanobacteria. Additionally, when applied together, some components in the mixture, though not toxic themselves, could be acting synergistically with the known toxins, possibly by enhancing their uptake and thus increasing the toxic response. This was not the only instance of such observations being reported. For example, extracts from a *Planktothrix* bloom were tested for embryonal and larval toxicity in zebrafish [99], and the induced toxic effects, interestingly, did not correlate with the microcystin content of the extracts. The most pronounced toxic effects were mainly induced by methanol and aqueous extracts, which significantly affected embryonic development, causing malformations, and also influenced hatching rates and caused premature hatching and growth delays. Reports concerning the developmental toxicity of MC-LR after direct submersion of zebrafish embryos are conflicting, and while some of the first experimental results provided very little evidence of such effects (although growth and survival rates were reduced after exposure to as little as 5 µg per liter) [86], more recent studies have demonstrated the occurrence of significant developmental defects in the exposed individuals [100].

While exploring the possibility of parental transmission of MC-LR toxicity in zebrafish, a significant drop was reported in the expression levels of growth genes (GH, GHra, GHrb, IGF1, IGF1ra, and IGF1rb), as well as a significant decrease in the transcriptional levels of immune-related genes IFN-1 and TNF- in all untreated F1 experimental groups [101]. These results indicated growth inhibition and immune suppression effects translated from the exposed parental groups to their offspring. An MC-LR-induced change in body weight and length in offspring of the exposed zebrafish was also observed, which correlates with the registered changes in the expression levels of growth genes. In recent years, the impact of MCs on endocrine systems has become a growing concern. Studies involving zebrafish show that both pure toxins and bloom samples of the cyanobacterium *Microcystis* have endocrine-disruptive potential and, therefore, represent a potential risk to human health and the environment. Changes in vitellogenin (vtg) levels have been identified as a sensitive biomarker for detecting the environmental concentrations of endocrine-disrupting chemicals in aquatic vertebrates. Vitellogenin is a phospholipoglycoprotein whose synthesis in the liver is influenced by the steroid hormone estradiol and is utilized by growing oocytes in generating yolk proteins during vitellogenesis [102]. Eight functional *vtg* genes are confirmed in the zebrafish genome [103], among which *vtg1* is thought to be the most highly expressed and is most commonly used in cyanobacterial toxicity testing. A modulation of the endocrine system function and disruption of oogenesis in female zebrafish exposed to 2, 10, and 50 µg/L of MC-LR for 21 days were reported [104]. Both the transcription level of liver *vtg1* (along with other factors regulating oogenesis) and plasma levels of the vtg hormone were affected by the toxin in a dose-dependent manner, increasing in individuals exposed to 10 µg/L of MC-LR and decreasing when a higher concentration (50 µg/L) was applied. Additionally, MC-LR exposure resulted in the decreased success of fertilization and hatching in female zebrafish from both treatment groups, which is indicative of trans-generational effects of microcystin. Another recognized effect of MCs in fish is the modulation of primary stress responses, which are regulated by the hypothalamic-pituitary-adrenal axis responsible for the release of corticosteroid hormones into circulation [105]. The changes in the expression of several genes within this system, including the corticotropin-releasing factor (CRF), thyroid-stimulating hormone (TSH), sodium/iodide symporter (NIS), thyroid peroxidase (TPO), transthyretin (TTR), thyroglobulin (TG), thyroid receptors (TRα and TRβ), and iodothyronine deiodinases (Dio1 and Dio2) have been successfully used in detecting MC-LR-induced thyroid disruption [106,107,108,109].

Additionally, since the nervous system of zebrafish has shown considerable similarity to that of mammals, this organism could potentially serve as a suitable model for the analysis of neuronal development and function defects [110]. For this reason, efforts have been made in identifying genes in zebrafish that can be used as biomarkers for the developmental neurotoxicity of cyanobacteria. A toxic bloom of *M. aeruginosa* was found to have a detrimental impact on the neuronal development of zebrafish, influencing both dopaminergic and cholinergic systems and altering the transcription levels of genes involved in the normal functioning of the nervous system [111]. Among the observed genes (*elavl3*, *shha*, *nestin*, *gap43*, *nkx2.2a*, and *ngn1*), significantly lower expression levels were observed for *elavl3*, an accepted early neuronal biomarker, in response to all of the used cyanobacterial concentrations (as low as 0.02 optical density (OD) of cyanobacterial cells), while the highest concentration (0.08 OD) significantly down-regulated all of the tested genes. Moreover, after exposure to a toxic *M. aeruginosa* bloom sample, both acetylcholinesterase (AChE) and dopamine levels in zebrafish larvae declined in a dose-dependent manner, with the most significant changes found in the group treated with the highest algal concentration of 0.08 OD. Similar targets among genes associated with neurotransmitter systems were chosen [112] (*α1-tubulin*, *manf*, *mbp*, *gap43*, *gfap*, *shha*, *neurogenin*, *nestin*, *nr4a2b*, *chrna7*, and *ache*) when the developmental toxicity of MC-LR was examined. Authors reported a significant down-regulation in *α1-tubulin*, *manf*, and *shha* caused by 3.2 mg/L of the toxin, while the expression levels of *mbp*, *chrna7*, *gap43*, and *ache* genes were significantly elevated after exposure to 1.6 mg/L MC-LR. The transcription levels of *gfap, neurogenin*, and *nr4a2b* remained unchanged. Brain acetylcholinesterase (AChE) activity in adult zebrafish was analyzed after exposure to different MC-LR concentrations [113]. A significant increase in the AChE activity was observed after 24 h exposure to the toxin dissolved in water at a concentration of 100 µg/L (31.14 ± 1.39; 27%; pb 0.05) compared with that of the untreated control group (24.50 ± 1.80). However, direct interference of MC-LR with the enzyme was rejected, as no significant changes were detected when testing the toxin in the enzyme assays directly. Nonetheless, these findings have provided evidence that brain AChE may be another potential target of this toxin. In several of these cases, reported neurological changes have been connected with behavioral anomalies which were also observed in the previously exposed individuals. For example, MC-LR has been shown to affect swimming behavior in exposed zebrafish. An increase in the motility of *D. rerio* was reported when exposed to low concentrations of MC-LR (0.5 µg), while exposure to higher concentrations (50 µg) resulted in a significant decrease in their motility [114]. The initial effect caused by lower doses of the toxin could merely be a reaction to the stimulus present in the environment, which turned into immobilization when the concentration was elevated. The effects of MC-LR exposure on zebrafish swimming activity were also evaluated in a behavioral test [115], where it was observed that MC-LR at a concentration of 100 µg/L decreased the distance traveled by 63% and increased the immobility time by a factor of three when compared with the control group. Additionally, a significant increase (around 93%) in the time spent at the bottom portion of the tank was observed when exposing zebrafish to 50 and 100 µg/L of the toxin, which is suggestive of an anoxygenic effect.

The effects of other cyanobacterial toxins, besides microcystins, have been successfully described using zebrafish as model organisms. The developmental effects of STX in zebrafish larvae directly exposed to the toxin dissolved in the aqueous media were described [116]. Exposure of the embryos to 481 ± 40 µg STX equivalents/L resulted in morphological abnormalities including severe swelling of the eyes, pericardium, and yolk sac, as well as craniofacial deformities. Additionally, sensorimotor deficits were observed, with total paralysis manifested with increased duration of exposure, to which more developed embryos were particularly sensitive. Most of the sublethal effects, including paralysis and, to some degree, the physiological abnormalities, seemed to be reversible by transferring the larvae to clean water, which is consistent with the described mechanism of action of STX. The effects of two *Cylindrospermopsis raciborskii* strains were tested in chronic and acute experiments with zebrafish larvae and adults [117]. Acute testing showed no effect of either strain on the survival of adult zebrafish; however, both strains negatively affected the survival of zebrafish in the larval stage of development, causing up to 40% lethality after four days, and 100% by the end of the test (7 days).

#### 2.2.2. Oxidative Stress Induction—*D. rerio*

Most of the literature investigating the oxidative stress response in zebrafish has focused on the effects of microcystin exposure on these organisms. The ability of microcystins to induce oxidative stress or alter the activity of vital components of the antioxidant system in aquatic organisms has been demonstrated in adult and developing zebrafish. The available research results indicate that the antioxidant response of these organisms to MCs mainly depends upon the dose of the toxin they are exposed to. A significant increase in sGST and mGST activities was reported in *Danio rerio* exposed to 0.1–2.0 µg/L MC-LR, while increasing the dose of the toxin seemed to suppress the GST activity [118]. As the observed response was dose-dependent, it was suggested that this defense system in embryos was not able to cope with MC-LR concentrations higher than 2.0 µg/L. This is consistent with other reports indicating that higher concentrations of MCs inhibited GST activity in zebrafish [119,120]. Interestingly, available research results reveal that the activities of other enzymatic components of the antioxidant system tend to decrease in response to MC exposure as well. For example, the activities of catalase (CAT) and glutathione-peroxidase (GPx) decreased significantly under different concentrations of MC-LR (3 and 30 µg/L) [121] while, in the same work, exposure to lower doses of MC-LR significantly increased GST activity. Lower microcystin concentrations have been shown to induce the activity of other antioxidants as well. Enzyme activities of GPx and superoxide-dismutase (SOD) increased at lower doses (≤1 µg/L) and decreased at higher concentrations (≥5.0 µg/L) in several different organs (including the liver) of adult zebrafish after MC-LR exposure [119]. Exposing zebrafish to intact cyanobacterial bloom samples, instead of purified toxins, resulted in similar alterations of enzyme activities. The toxicological effects of bloom samples of a microcystin-producing *Microcystis* strain and a cylindrospermopsin-producing *Oscillatoria* strain, with cell densities ranging from 0.5 to 2 × 10^6^ cells/mL, were evaluated in zebrafish embryos. In the groups exposed to the *Microcystis* samples, a significant increase in the activities of CAT, SOD, and GPx enzymes was observed for the treatments of up to 10^6^ cells/mL, while the highest-used concentration (2 × 10^6^ cells/mL) induced a significant decrease in the activity of these enzymes. On the other hand, the activities of all enzymes increased significantly in response to the *Oscillatoria* strain samples in all of the applied concentrations. The activity of CAT has also been shown to decrease in zebrafish juveniles whose parents were exposed to as low as 5 µg/L of MC-LR [101], which demonstrates the possibility that the oxidative stress promotion caused by MCs may be transferred to the exposed organisms’ offspring. Additionally, the activities of SOD and GPx were significantly decreased in F1 juveniles obtained from parents from all the treatment groups (1, 5, and 20 µg/L MC-LR). These observations mainly underline the significance of the applied concentration in MC-LR-induced alteration of antioxidative enzyme activity.

It is also important to note that the alterations caused by MCs leading to the antioxidative enzyme response can also be observed at the gene expression level. The mRNA expression levels of *cat1*, *gpx1a*, and *gstr1* genes were analyzed in the liver of adult zebrafish exposed to MC-LR [121]. Results showed a significant inhibition of the tested genes, indicating that CAT, GPx, and GST activities were all suppressed at this level following the treatment with MC-LR. The toxic effects of MC-LR on the reproductive system of zebrafish were evaluated through the induction of oxidative stress by observing the changes in the gene expression of important antioxidant enzymes after intraperitoneal injection with 50 and 200 µg MC-LR/kg of body weight [122]. For both treatment groups, the initial increase in the mRNA expression of CAT1, SOD1, and GPx1a was followed by a significant decrease within 48 hpi (hours post injection) compared with the control, while the expression of GSTr1 progressively decreased during the same period. It is important to add that, at the end of the experiment (168 hpi), the transcriptional levels of all the tested genes returned to levels similar to those recorded in the control group.

Recently, zebrafish have gained increasing popularity in the study of cyanobacterial toxicity due to their unique ability to serve as a model for observing sublethal effects resulting from both acute and chronic exposure, which is critical in identifying potential hazards and developing protection strategies. By combining the scalability and efficiency of in vitro tests with the physiological complexity of in vivo models, zebrafish are well-suited for identifying potential toxic effects of cyanobacterial metabolites. Zebrafish toxicity assays offer a promising way to streamline toxicity testing timelines, use smaller volumes of test samples, and reduce unnecessary costs associated with mammalian studies. The available findings suggest that zebrafish are effective not only in detecting various cyanobacterial metabolites, including those with cytotoxic, genotoxic, and neurotoxic properties, as well as those with the potential for endocrine disruption in aquatic environments, but also in assessing and quantifying the toxic effects of these metabolites and establishing the primary mechanisms of action in vertebrate systems. While zebrafish hold promise as a model for toxicity testing, it is important to acknowledge that as a non-mammalian model, the relevance of data obtained from zebrafish studies to mammalian systems remains partly unclear. Furthermore, the uptake and metabolism of cyanobacterial toxic compounds in zebrafish are not yet well-characterized, and the extent to which the chorion barrier may impact uptake from the surrounding medium is still uncertain. These considerations underscore the need for further investigation to fully understand the limitations and potential of zebrafish toxicity assays in the context of broader toxicity testing programs.

### 2.3. Oryzias Latipes Bioassay

Medaka fish (*Oryzias latipes*) are small freshwater fish that have become increasingly popular as model organisms in toxicity testing (Table 1). This aquatic animal shares many of the characteristics considered valuable in toxicity testing with zebrafish. Like zebrafish, medaka are transparent during early development, which makes them ideal for studying the effects of toxic substances on organ development and function [123]. Medaka also have a short life cycle, with their eggs hatching within a week and their full development taking only about two months, which allows for faster testing and evaluation of toxic developmental effects [124,125]. Another benefit of using medaka fish in toxicity testing is their ease of maintenance and low cost compared with other vertebrate models. They are hardy and can be kept in small tanks, making them an attractive option for researchers working in limited lab spaces [123]. Additionally, the medaka genome was fully sequenced in 2007, providing a powerful tool for genetic studies and helping to understand the mechanisms of toxicity at the molecular level [124]. The studies utilizing medaka in the investigation of cyanobacterial toxicity have primarily focused on histopathological changes resulting from acute and chronic exposure, as well as the molecular and reproductive effects underlying these changes [125,126,127]. A study was conducted to investigate the toxicity of extracts obtained from two variants of *Planktothrix agardhii*, a microcystin-producing (PMC 75.02) and an MC-free strain (PMC 87-02), in adult medaka fish [127]. Extracts of natural bloom samples containing several MC variants were also tested. The treatments were administered using gavage, and test organisms were observed for signs of physiological stress and pathological changes. The study found that the mortality rate was significantly higher in fish groups treated with the extract of the MC strain PMC 75.02 (45.5%) compared that of with both the MC-free strain PMC 87.02 (3.3%) and the pure MC-LR sample (18.7%). Furthermore, severe damage to the liver and intestine of the fish was observed following exposure to PMC 75.02 and *P. agardhii* bloom extracts, while the MC-free strain had no effect on these tissues. The liver of the fish exposed to the toxic extracts showed areas of lysed cells, disconnected hepatocytes, depleted glycogen and glycoprotein storage, and lipid vesicles. The intestine also showed similar damage, with isolated enterocytes and lysed cells. Exposure to MC-LR was also shown to have significant effects on the digestive and associated systems during early (embryo-larval) development of medaka, including many of the previously mentioned liver defects. Chronic exposure to MC-LR and *M. aeruginosa* extracts were also shown to result in hepatocyte lysis and a decrease in intrahepatocyte glycogen reserves [128]. Similar changes in liver function were described after chronic exposure via balneation to MC-LR, as evidenced by the higher level of activity of liver enzymes and histological changes, along with spleen and intestine defects [126]. Furthermore, this was the first report of the reproductive effects of MC-LR in medaka, as the treatment also caused pathological modifications in the gonads of both male and female medaka fish. In female gonads, the modifications included a reduction in the vitellus storage, lysis of the gonadosomatic tissue, and disruption of the relationships between the follicular cells and the oocytes and disrupted spermatogenesis in males. These effects were further explored with the aim to investigate the potential reproductive toxicity of microcystin, focusing on hepatic alterations in medaka fish chronically exposed to MC-LR and *Microcystis aeruginosa* PCC 7820 extract [129]. Both pure toxin and *Microcystis* extract had adverse effects on reproductive parameters, including fecundity and egg hatchability, with significant decreases observed in these parameters under all MC-containing treatments. Histological, proteome, and transcriptome analyses revealed glycogen storage loss and cellular damage in the livers of toxin-treated female fish, as well as dysregulation of hepatic proteins, including a notable decrease in quantities of vitellogenin and choriogenin. These findings suggest a modification in liver glycogen synthesis or consumption processes and a depletion of hepatic glycogen, which could reflect increased energy requirements for organism homeostasis, tissue repair, and molecular detoxification processes. Chronic (8 weeks) dietary exposure to MC-LR inhibited growth, decreased survival in embryos, and lowered the RNA/DNA ratio of whole fish [130]. It was also shown that the effects in fish were gender-specific, with females being more affected by the treatments than males. This observation was substantiated using proteomic and metabolomic analyses of medaka livers following 96 h exposure to monoclonal cultures of *M. aeruginosa* [131]. However, at this level of analysis, deleterious effects in fish exposed to non-MC-producing strains were also reported, indicating that metabolites other than microcystins can induce a response in the exposed medaka, even though MC-producing strains caused more severe changes.

**Table 1 biology-12-00711-t001:** Summary of experiments using the most common vertebrate biotests in cyanobacterial toxicity testing.

Model Organism	Observed Parameter	Sample Type	Exposure Duration	Effective Concentration of the Agent	Main Observed Effect	Reference
Mouse	Mortality rates	MC-LR containing bloom samples—i.p. injection	24 h	LC50 = 22–250 mg/L dry weight (dw)	Lethality	[132]
	DNA damage	Purified MC-LR—i.p. injection and oral exposure	24 h	2 and 4 mg/kg dw	DNA lesions induced in the liver, kidney, intestine, and colon	[133]
	Lung damage after chronic exposure	Purified MC-LR—oral exposure	12 months	5–40 µg/L	Lung impairment—thickening of the alveolar septa	[134]
	Clinical changes, hystopathological changes, serum indicators of hepatic toxicity, and general homeostasis	Purified MC-LA—oral exposure	24 h	3 mg/kg	Weight loss; elevated liver/body weight; liver score; serum levels of ALT, AST, GLDH; BUN/creatinine ratios and total serum bilirubin; reduced serum glucose	[59]
		Purified MC-LR—oral exposure		5 mg/kg	Weight loss; elevated liver/body weight; serum levels of ALT, AST, GLDH; liver score; reduced serum glucose	
		Purified MC-LY—oral exposure		5 mg/kg	Weight loss; elevated liver/body weight; serum levels of ALT, AST, GLDH; liver score; reduced serum glucose	
		Purified MC-RR—oral exposure		22 mg/kg	Weight loss, reduced serum glucose	
		Purified MC-YR—oral exposure		7 mg/kg	Weight loss, elevated liver score, BUN/creatinine ratios, reduced serum glucose	
	Chronic exposure reproductive effects	Purified MC-LR—oral exposure	6 months	30–120 µg/L	Testis structure loss, cell abscission and blood–testis barrier (BTB) damage	[135]
			12 months	1–120 µg/L		
	Histopathological effects	Purified ANTX-a—i.p. injection	15 days	0.5–1 μg/l	Fatty liver degeneration, congestion, inflammation and necrosis, morphological kidney alterations, testis structure loss, and decrease in the number of elongated spermatids	[136]
	Survivorship	Bloom samples—i.p. injection	24 h	LC50 = 445.45 mg dw/kg	Mortality	[137]
	Survivorship	Bloom samples—i.p. injection	24 h	LC50 = 20–908 mg dw/kg	Mortality	[138]
Medaka fish (*Oryzias latipes*)	Survival and developmental toxicity	Purified MC-LR	10 days	1–10 µg/mL	Up to 90% reduction in survival rates and altered hatching rate	[139]
	Survival and developmental toxicity	*Microcystis* laboratory culture	5 days	13–46 × 10^6^ cells/mL	Decreased heart rate	[140]
		Bloom samples	15 days	56.3–244 × 10^6^ cells/mL	Up to 100% reduction in survival rates, altered hatching rate, reduced body length, yolk sac edema, decreased heart rate	
	Toxicity after oral exposure	Purified ANTX-a	10 days	LC50 = 11.5 µg/g	Decreased survival at higher doses, no accumulation in tissues, and recovery after 24 h of exposure to lower doses	[141]
	Reproductive toxicity after chronic exposure	Purified MC-LR	28 days	1–5 µg/L	Liver glycogen storage loss and cellular damages, altered fecundity and hatching rate, induction of circadian-rhythm-related genes	[129]
		Crude extracts of *M. aeruginosa* PCC7806				
	Survival and developmental toxicity	*P. agardhii* extract	11 days	10–50 × 10^−3^ mg/mL	Up to 81% reduction in survival rates, hepatobiliary abnormalities, altered hatching rate	[142]
	Chronic exposure effects	Purified MC-LR	30 days	5 µg/L	Histopathological modifications of the female and male gonads	[126]
Zebrafish (*Danio rerio*)	Survival and developmental toxicity	Purified CYN—direct immersion	5 days	Up to 50 µg/mL	No adverse effects observed	[96]
		Purified CYN—microinjection		LC50 = 4.5 fmol/embryo	Up to ~15% reduction in survival rates	
	Survival and developmental toxicity	Purified CYN	96 hpf	200–2000 nM	Reduction in survival rates, decreased heart rate	[143]
				20–2000 nM	Reduction in body length, reduced eye size, pericardial edema, curved spine, tail deformity, uninflated swim bladder, altered hatching rate	
	Survival and developmental toxicity	Purified MC-LR—microinjection	48 hpf	300–900 nM	Up to 82.5% decrease in survival rates, tail deformity, pericardial edema, blastomere coherence inhibition	[100]
	Survival and developmental toxicity	Purified CYN	120 hpf	500–2000 µg/L	Up to 40% decrease in survival rates, reduction in body length	[144]
				10–2000 µg/L	Pericardial edema, yolk sac edema, swim bladder abnormalities	
	Survival and developmental toxicity	Purified STX	7 days	481 ± 40 µg/L	Edema of the eyes, pericardial and yolk sac edema, swim bladder abnormalities, craniofacial deformities, decreased mobility	[116]
	Survival and developmental toxicity	Crude extracts	48 h	30–71 µg/mL dw	Decreased survival and developmental malformations	[145]
		Crude extracts of *M. aeruginosa* PCC7806		12 µg/mL dw		
	Enzyme activity alteration	Purified MC-LR—direct immersion.	24 h	100 μg/L	Increased AChE activity (27%), increased *ache* gene expression (17%)	[115]
		Purified MC-LR—microinjection			No effect on AChE acitivity	
	Developmental toxicity	Purified MC-LR	30 days	5 and 20 µg/L	Dose-dependent reduction in SOD, CAT, and GPx activities	[101]

## 3. Bioassays with Invertebrate Animal Models

In recent years, the interest in using invertebrate models for toxicity testing of cyanobacterial metabolites has been growing, as they offer several advantages over traditional mammalian models, including cost-effectiveness, high throughput, and fewer ethical concerns [81,146,147]. Moreover, invertebrate models can provide valuable insights into the sublethal effects of cyanobacterial toxins on the environment and ecosystem. In this chapter, we review the current state of knowledge on invertebrate toxicity testing of cyanobacterial toxicity, focusing on the strengths and limitations of various invertebrate models and their suitability for different types of toxicity assessments and samples.

### 3.1. Artemia Salina Bioassay

Brine shrimp lethality bioassay [148,149,150] is a fast and inexpensive test with no culture maintenance required, as *A. salina* cysts (eggs), which are commercially available, can be stored for several years at −20 °C once freeze-dried, readily hatched from this state within 24 h into nauplii (larvae), and used in experiments without the need for special equipment. Uniformity of the experimental groups can be ensured by using larvae in the same developmental stage, as well as by taking into consideration their geographical origin, as these factors can influence growth, reproduction, and survival rates. Testing procedures which employ a high-throughput approach have been described in the past [148] and have enabled rapid analysis of a large number of samples and dilutions in a single plate. This is convenient, as this assay has mostly been established as a rapid and inexpensive substitute for cytotoxicity assays, especially valuable in laboratories not equipped with cell culture facilities. Tests can be conducted by introducing neonates into microtiter plates containing artificial salt water and the test solution and incubating the plates at 30 °C under illumination for 48 h [150,151]. The main response criterion is mortality (lack of motility), which is recorded via microscopic examination after 24 and 48 h of exposure and from which LC50 values are calculated. Studies comparing the successfulness of the *Artemia* bioassay with that of the mouse bioassay in testing toxic cyanobacterial metabolites have shown great similarity and a good correlation between the obtained results, confirming the reliability of this test for the investigation of cyanobacterial toxicity [152,153,154]. However, research data on the correlation between the sensitivity of the brine shrimp assay and some tumor cell lines in determining cytotoxic potential have been conflicting and will be discussed in the following chapter. One of the recognized downsides of the test is the decreased solubility and bioavailability of some substances in a saline medium, which is necessary for the normal functioning of brine shrimp [153,154]. Additionally, some inconsistencies could potentially arise when conducting tests using toxins associated with freshwater cyanobacteria in a saline environment [155], though such issues have not yet been reported.

#### 3.1.1. Whole-Organism Responses—*A. salina*

This assay has proven to be useful in testing the extracts from both hepatotoxic [156] and neurotoxic cyanobacterial isolates [151,157] and has shown a good correlation between the toxin abundance in samples and the observed mortality rates. It has since been extensively used for the purpose of investigating the biological activity of cyanobacteria, although predominantly as a rapid screening procedure for toxic compounds in large sample groups [158,159]. Due to its simplicity and low cost, the brine shrimp assay continues to be used in preliminary cytotoxicity testing today. However, several studies have shown a discrepancy when comparing this assay and other techniques and testing models, including mammalian cells [159,160]. There were discrepancies when the biological activity of 86 extracts, obtained from 43 samples of freshwater and terrestrial cyanobacteria, was investigated using the brine shrimp lethality assay, as well as the KB cell line (ATCC CCL 17; human nasopharyngeal carcinoma) and the Caco-2 cell line (ATCC HTB-37, human colon adenocarcinoma) [160]. Seven extracts (8.1%), five hydrophilic and two lipophilic, were found to be highly active (lethality ≥ 60%) against *A. salina* nauplii at 500 *ppm* and no correlation was discovered between brine shrimp lethality and cytotoxic effects observed in the case of the tested cell lines, with the exception of two extracts which were active against both Caco-2 cells and brine shrimp. In another paper [161], the bioactivity of a total of 80 cyanobacterial extracts of cultured freshwater and terrestrial cyanobacteria was investigated using the *Artemia* assay, and 26% of the extracts exhibited a significant lethal effect (lethality > 70%) against brine shrimp. None of the extracts were, however, deemed toxic to KB cells and only one strain showed weak activity toward T-24 cells (human bladder carcinoma), recorded as IC50 = 16.1 pg mL^−1^. A further work [162] also reported no correlation between the cytotoxicity against KB cells and the observed *A. salina* mortality after the bioactivity testing of 44 cyanobacterial extracts, of which 38.6% exhibited cytotoxicity against KB cells, while only two extracts significantly affected brine shrimp.

This discrepancy could be a consequence of cyanobacterial metabolites mainly targeting the metabolic pathways present in mammalian cells [163], which would make the *A. salina* test a suboptimal solution when screening for cytotoxicity. However, the brine shrimp assay was found useful when testing the effects of cylindrospermopsin, which is, according to its described primary mode of action, a cytotoxic agent. The extracts and purified CYN of *C. raciborskii* were shown to be toxic to *A. salina* [164]. Additionally, the same bioassay was used to compare the toxicity of three protein synthesis inhibitors (chloramphenicol, cycloheximide, and tetracycline) to that of cylindrospermopsin. A decrease in the LC50 values was observed with increased exposure duration. The highest concentration of purified CYN (20 µg/mL) applied in the experiment resulted in 100% mortality after 40 h of exposure, while the extracts of the CYN-producing strain caused the same effect at a concentration of 10 mg/mL after 24 h, at 2 mg/mL after 40 h of exposure. When the protein synthesis inhibitors were compared, the ones interfering with eukaryotic protein synthesis (CYN and cycloheximide) were more toxic to *A. salina*, with CYN being the most toxic of the three tested agents.

The bioassay with *Artemia* nauplii has also been a valuable tool for the detection and monitoring of toxigenic cyanobacteria in water supplies (Table 2). There have been several reports of its successful use in identifying the toxic properties of water samples [148,165,166]. Following an *Aphanizomenon ovalisporum* bloom in a freshwater reservoir in Andalusia-Spain, *A. salina* bioassay was used to investigate the toxicity of the samples [167]. The toxicity values, expressed as LC50 Chl a (i.e., the amount of chlorophyll a contained in the extract that can cause 50% mortality of the test organisms), ranged from 2.8 to 3.4 µg. The analysis, conducted using high-performance liquid chromatography (HPLC), revealed no microcystin or anatoxin-a; however, the presence of CYN in the cyanobacterial sample was confirmed, which could have been the causative agent for the observed toxicity. Furthermore, *A. salina* bioassay was utilized to determine the toxicity of the intra- and extracellular contents of *C. raciborskii* blooms samples following a significant fish mortality event in Aleksandrovac Lake, Serbia [168]. After ruling out heavy metal poisoning, microbiological agents, waste discharge, and other physicochemical factors, as well as the presence of common cyanotoxins (CYN, MCs, and STX), the strong reaction in the bioassay suggested the presence of uncharacterized toxic agents within the cyanobacterium *C. raciborskii*.

#### 3.1.2. Mediation of Cyanobacterial Toxicity—*A. salina*

*Artemia salina* have been of great value in the detection of the toxicity of microcystins. The biological activity of MC-LR was tested against several invertebrate species, and a 24 h LC50 value of 3.75 µg/mL was obtained using the brine shrimp assay [169]. Results in the same range were reported later, when toxicities of cyanobacterial bloom samples were evaluated based on their lethality to brine shrimp [170]. Both pure microcystin-LR standard and the tested cyanobacterial bloom extracts were found to be toxic to *A. salina* in a dose-dependent manner, with calculated LC50 values ranging from ~1 to 40 mg dry weight of cells per mL for the hepatotoxic bloom samples and 6.80 µg/mL for the purified toxin. The mediation pathways of MCs and other cyanotoxins have also been investigated using the brine shrimp assay in the past. Conjugation with glutathione (GSH), which is catalyzed by glutathione-S-transferases (GST), has been recognized as one of the most important steps in the metabolism of microcystins, as well as the primary route of MC detoxification in aquatic organisms [171]. During this process, toxic compounds are transformed to more polar metabolites, which are excreted from the organism more easily [172]. Differential expression of several GST isozymes was measured at different developmental stages of *Artemia salina* (cysts, 24 h old nauplii, and adult individuals) in response to MC-LR, MCHtyR, and NODLN [173]. A significant elevation was observed in the activities of both microsomal (mGST) and solubile (sGST) fractions after exposure to the cyanobacterial toxins. All three of the tested toxins were conjugated to glutathione via GST as an initial step in their detoxication. A similar increase in GST enzyme activity, induced by the exposure to microcystins, was observed in other shrimp models such as *Palaemonetes argentinus* and *Litopenaeus vannamei* [174,175], which indicates that this pathway could be involved in the elimination of MC and NODLN in these organisms. It was also observed that the activities of both GST fractions (mGST and sGST) significantly increased in *A. salina* in response to MC-LR exposure [98]. Additionally, antioxidants seem to be able to reduce MC-LR toxicity [176]. This study was based on a hypothesis that a 4 h pretreatment with antioxidants (either vitamin E or Trolox) could lessen the harmful effects of MC-LR exposure in *A. franciscana*. Pretreatment with both antioxidants resulted in significantly reduced mortalities (approximately 50%) in the test organisms exposed to 40 µg/mL MC-LR. Pre-exposure to cyanobacterial LPS’s was also proven to protect aquatic invertebrates against some of the purified toxins to a degree. This effect was observed in *A. salina* (along with *Daphnia magna* and *D. galeata*), pretreated with purified cyanobacterial LPS (2 ng/mL) and exposed to MC-LR and CYN in the concentration range of 1 pg–20 µg/L [98]. This 24 h pre-incubation with LPS altered the LC50 values recorded for both toxins, though the difference was much more pronounced in the case of MC-LR, of which the initial LC50 value for *A. salina* was 2 µg/mL, which increased to 8 µg/mL with LPS pre-exposure.

The advantages of using *Artemia salina* cyanotoxicity testing are numerous. First, it is a fast and inexpensive way to screen a large number of water samples, making it ideal for initial screening purposes. Additionally, *Artemia* have a particular sensitivity to the presence of certain cyanotoxins such as MC and CYN, making them an effective tool for detecting these toxins. They are also easy to maintain and handle due to their small size and simple requirements. These features make it an ideal organism for laboratory use, especially in resource-limited settings. However, one of the main limitations is that not a lot of information can be gained from it compared with other models, as mortality (lack of motility) is one of the only endpoints tested, while some protocols also observe erratic (unusual) swimming. This can limit the scope of the research and the depth of the information that can be obtained. Another potential disadvantage of using *A. salina* is its sensitivity to other cyanobacterial metabolites beyond MC and CYN. This can lead to false negatives with certain toxins. Additionally, according to multiple sources mentioned previously, there is a low correlation between *A. salina* testing results and cell line testing results, which may limit its usefulness in certain research contexts.

### 3.2. Daphnia sp. Bioassays

Species of the genus *Daphnia* (Müller) are ubiquitous in temperate freshwaters and represent important indicators of changes in various aquatic environments. Within these habitats, they are often the primary grazers of algae, bacteria, and protozoa and, as such, represent an integral ecological component [177]. Their application in toxicological studies started as early as 1920s and has become a widely implemented practice when it comes to the environmental monitoring of pollutants with the adoption of guidelines concerned with acute [178] and chronic [179] toxicity testing. Among various characteristics that contribute to them being valuable model organisms in toxicity studies, including fast reproduction, large clutch size, ease of culture maintenance, and high sensitivity to the presence of environmental contaminants, is the ability of clonal reproduction [180]. Because of this trait, it is possible to achieve and maintain genetic uniformity in cultures, providing a stable, constant genetic background to which experimental results can be compared. This aspect has been further potentiated by the development of *Daphnia* genome [181]. Immobilization is the most commonly used endpoint in toxicity assays for these species, but they have also shown a particular sensitivity to sub-lethal concentrations of harmful substances. This sensitivity enables early detection and monitoring of toxin-induced changes during intoxication [182].

#### 3.2.1. Whole-Organism Responses—*Daphnia*

The impact of toxic cyanobacteria, especially strains of the genus *Microcystis*, on the survivability of daphnids has been well-studied [183,184,185,186]. To obtain a more accurate understanding of the role of microcystins in daphnid poisoning, one study [187] utilized a mutant clone of *M. aeruginosa* that had a mutation in a microcystin synthetase gene, rendering it incapable of producing the toxin. This mutant clone was compared with the original toxin-producing strain, after which it was possible to confidently attribute the observed toxic effects in daphnids to microcystins. In addition to significantly increased mortality rates in daphnids exposed to the toxic clone, a distinct reduction in swimming activity was also reported in the affected animals. Furthermore, changes in gene expression in *D. magna* exposed to a toxic wild strain of *M. aeruginosa* as well as to a non-toxic cyanobacteria *Synechococcus elongatus,* were analyzed with the aim of finding out whether the detected changes were merely a response to other non-toxic cyanobacterial metabolites [188]. The expression levels of two of the genes selected in the experiment (*GapDH* and *UBC*) were considerably lower after the exposure of test organisms to *S. elongatus* compared with those recorded in the *M. aeruginosa* group, which implies a toxin-induced effect on the processes of glycolysis and protein catabolism.

Ingestion of microcystin-producing cyanobacterial cells was described as the primary mechanism of intoxication in *Daphnia* [189]. Digestion of *Microcystis* cells releases microcystins which accumulate in the midgut cavity and may be transported directly into the bloodstream. This was to be expected, as daphnids are unselective filter feeders and are exposed to toxic cyanobacteria in their natural environment. However, this finding directly connected feeding activity with the survival of exposed daphnia. It is therefore a logical assumption that the mechanism of intoxication with other cyanobacterial toxins also involves digestive uptake. A decreased feeding rate was reported in *Daphnia carinata* exposed to *Aphanizomenon flos-aquae* filtrate and purified STX, measured by the beating frequency of thoracic appendages [190]. Additionally, an increased rate of post-abdominal rejection of accumulated material was observed, possibly due to the presence of toxic compounds present in the medium. Animals were able to recover to their pre-treatment levels of activity much faster after exposure to pure STX than to the filtrate. A strong inhibition of *D. pulex* occurred when the organism was exposed to the filaments of *Anabaena flos-aquae*, *A. affinis*, and to the purified anatoxin-a [191]. The test organism responded similarly to *A. flos-aquae* filaments and the purified toxin, which means that the filamentous form of *Anabaena* was not responsible for the observed effects. It was concluded in the same experiment that the susceptibility of daphnids increased in higher temperature conditions, which could be an indication of temperature influencing microcystin intake in *Daphnia*. Two strains of the cyanobacterial species *Cylindrospermopsis raciborskii* (T2 and T3) had toxic effects in *Daphnia similis* and *Ceriodaphnia dubia* used as model organisms in acute and chronic toxicity tests [129]. Both untreated cyanobacterial cultures caused the immobilization of *Daphnia similis* in the acute test, though they were much less toxic to the test organisms once filtered, which suggests that there is little to no expulsion of toxins from the cells and into the surrounding medium during cell growth, prior to cell lysis. Apart from survivorship, both strains caused a significant reduction in the production of neonates in *Ceriodaphnia dubia* chronic tests, with reported EC50 values as low as 2.651 cells/mL. Purified MC-LR toxin had an acute toxic effect on *D. pulex* after 24 h of exposure in another study [192], and their survivorship was lowered to approximately 80% at a toxin concentration of 1.66 mg/L and to 60% at a concentration of 3.32 mg/L. The LC50 value for *D. pulex* exposed to MC-LR is presumably found at even higher concentrations. In the same study, pure ANTX decreased the survivorship of daphnids to approximately 33% at a concentration of 1.66 mg/L, demonstrating a much stronger effect than the one registered in the case of MC-LR at the same concentration.

#### 3.2.2. Mediation of Cyanobacterial Toxicity—*Daphnia*

Some contradictory findings resulting from the investigation of daphnid interaction with toxic cyanobacteria could be explained by the capacity of *Daphnia* to develop tolerance to cyanobacterial toxins, which are often present in the natural environment. There have been indications of altered sensitivity of these animals to cyanotoxins, especially after they have been pre-exposed to certain cyanobacterial metabolites or cell constituents [98,193,194]. A significant effect on survivability, growth, and fecundity was observed in *D. magna* exposed to a *M. aeruginosa* strain [194]. Interestingly, the experiment also revealed that test organisms pre-exposed to toxic *M. aeruginosa* show increased tolerance (survival and growth rate) to the toxic effects, which implies that daphnids are capable of adapting to the presence of toxic *Microcystis* strains, given appropriate acclimation conditions. Exposing the parental generation of *D. magna* to microcystins for seven days led to an increase in GST activity in F1 offspring, which was in direct correlation with the MC-LR concentrations applied in the experiment (50 and 100 µg/L) [185]. A similar increase was observed in the activity of the enzymes catalase and malate dehydrogenase, which is indicative of a heightened capacity for oxidative stress mitigation and amplified energy production pathways. These results could indicate that MC-LR exposure can potentially heighten the tolerance of *Daphnia* offspring by indirectly influencing GST activity, which was indicated by the increased survival of juvenile individuals that originated from the exposed females. The first step of microcystin detoxification in *Daphnia* was proposed to involve conjugation with GSH, which is catalyzed by GST [195]. Increased activity of the glutathione S-transferase enzyme (GST) was also reported in another study [196] following 24 h exposure of *D. magna* to two strains of *Cylindrospermopsis raciborskii*. Since GST is involved in the biotransformation by catalyzing the conjugation of reduced glutathione to either xenobiotics or endogenous substrates, which usually results in detoxification, it is likely that this change was induced by the presence of certain toxic compounds produced by the cyanobacterial strain. This effect could be attributed to cylindrospermopsin, whose presence was confirmed using liquid chromatography–mass spectrometry (LC/MS/MS). This could mean that the cyanotoxin-related induction of GST enzymes is not specific to microcystin exposure but, rather, to all toxic cyanobacteria, or is simply a part of the general oxidative stress response. Further evidence on this topic was offered by exposing *Daphnia magna* to a diet of a toxic *M. aeruginosa* strain for 24 h [197] or one of the three different control groups, i.e., one group exposed to an MC-deficient mutant, another fed with the green algae *Chlamydomonas klinobasis*, and a third no-food group. The analysis showed significantly higher GST activity in the groups fed with the two cyanobacterial strains. However, no observable difference in the enzyme activity was reported between the groups fed with toxic and non-toxic *Microcystis* strains. In recent years, the use of OMIC approaches, including transcriptomics, proteomics, and metabolomics, has greatly enhanced our understanding of the sublethal effects of cyanotoxins on daphnids, as well as the described alterations in their sensitivity to these toxins [198,199,200,201]. In a recent transcriptomic study, the gene expression patterns of *Daphnia magna* were analyzed in response to a cyanobacterial strain that produces microcystins and its knock-out mutant [202]. The study identified transporter genes that are regulated by microcystins and likely contribute to *Daphnia*’s adaptation and tolerance to these toxic compounds, shedding light on the molecular mechanisms underlying microcystin tolerance in *Daphnia*. Through an analysis of proteomic profiles, it was found that both *D. magna* parents exposed to cell-bound microcystins and their neonates displayed distinct changes in protein abundance [201]. The parents exhibited a significant increase in proteins related to reproductive success, development, removal of superoxide radicals, and motor activity, while neonates showed a significant decrease in proteins related to apoptosis, metabolism, DNA damage repair, and immunity. Furthermore, in a study on the impact of *M. aeruginosa* on *D. magna*, it was found that dietary exposure significantly affected proteins related to lipid, carbohydrate, amino acid, and energy metabolism [200]. The study also revealed reduced growth rates in daphnids, leading the authors to suggest that this change in metabolism may be a compensatory mechanism.

Bioassays with *Daphnia* species have been of great use in cyanobacterial toxicity testing (Table 2). Besides the broad number of endpoints used for daphnids, in recent years, numerous studies have explored the alterations in gene expression induced by cyanotoxins in *Daphnia* [198,199,200,201,202], though the molecular mechanisms underlying their response to toxic cyanobacteria remain largely unresolved. Investigating gene expression and identifying novel biomarkers in ecologically relevant organisms, such as *Daphnia*, can provide valuable insights into environmental toxicity and offer information for ecological risk assessment. Moreover, gene expression profiling and determining toxicity modes can help to address adverse phenotypic outcomes linked to specific gene functions. Therefore, gene expression analyses can improve our understanding of the mechanisms by which cyanotoxins elicit or modulate adverse effects in daphnids, and they can help identify sensitive biomarkers responsive to cyanotoxicity. Connecting these findings with changes observable at higher levels of biological organization could provide a deeper understanding of cyanobacterial toxicity in the context of the entire organism. This understanding may also enable the detection of toxic effects at early stages in the toxic effects cascade.

### 3.3. Thamnocephalus Platyurus Bioassay

The commercially available biotest employing larvae of the freshwater anostracan crustacean *Thamnocephalus platyurus* (fairy shrimp) has successfully been used in the screening of cyanobacterial toxicity (Table 2), especially that of CYN and MC [203]. This is a cost-effective and standardized bioassay applicable to chemical substances, surface waters, wastewaters, groundwaters, aqueous extracts, and cyanotoxins. Thamnotoxkit F^®^ provides all necessary materials to conduct six 24 h mortality tests in a multiwell plate, using instar II-III larvae which are hatched from cysts. Guidelines provided in ISO 14380:2011 [204] outline a method for determining the lethal effects of toxicants on *Thamnocephalus platyurus* and a rapid test for sublethal effects after 1 h of exposure. A 60-min feeding inhibition test and 24-h mortality test are performed according to the Rapidtoxkit and Thamnotoxkit F^®^ standard operational procedures, respectively.

In an interlaboratory study conducted in 1997, Thamnotoxkit F^®^ microbiotest was evaluated for its potential as a tool in monitoring procedures and early detection of toxic blooms [205]. Extracts of five toxin-producing cyanobacteria (*Anabaena flos-aquae*, *Microcystis aeruginosa*, *Cylindrospermopsis raciborskii*, *Aphanizomenon flos-aquae*, and *Tychonema bourrellyi*) were selected for the assessment and the concentration series used ranged from 0.3 to 5.0 mg/mL. The results showed that *T. platiurus* larvae were most sensitive to the extracts of *Anabaena flos-aquae* and *M. aeruginosa*, which caused 100% mortality in almost all the concentrations used, except for the highest dilution, in which the mortality rate was still around 73.5%. The extracts of *C. raciborskii* caused 100% mortality starting from a concentration of 1.0 mg/mL, while *A. flos-aquae* caused total mortality at a concentration of 3.0 mg/mL. The usefulness of Thamnotoxkit F^®^ was demonstrated in detecting microcystin-LR in cyanobacterial bloom samples [206]. It was found to be more sensitive to the presence of this toxin than other assays, including the mouse bioassay. However, one of the main obstacles when implementing this assay into routine monitoring of bloom toxicity is the issue of hypersensitivity and lack of discriminative power when it comes to toxic and non-toxic cyanobacterial samples. The sensitivity and specificity of this and 16 other commonly used acute bioassays in the detection of microcystins in cyanobacterial samples were evaluated [207]. One of the main criteria in the assessment was the test’s ability to discriminate between samples with markedly different concentrations of MC-LR. Even though *T. platyurus* showed higher sensitivity to microcystins than any other applied model, mortality levels were significant even in the extracts without microcystins. This observation has also been substantiated by other studies reporting a lack of correlation between the assays’ response and the concentration of identified cyanotoxins in the tested samples [90,99].

### 3.4. Chironomus Bioassays

*Chironomus riparius*, also known as the harlequin fly or non-biting midge, is commonly used in toxicity testing procedures designed to assess the effects of pollutants and toxic substances potentially harmful to benthic organisms. It is known that toxic secondary metabolites released during cyanobacterial blooms in aquatic ecosystems can be harmful to benthic invertebrates [79] because after the bloom, dead cyanobacterial cells settle at the bottom of the basin and contaminate the benthic community, causing harm to organisms, including chironomid larvae, which are abundant in these environments. This insect species is sensitive to changes in water quality and provides a rapid, cost-effective, and sensitive model for evaluating the toxicity of water samples and a useful tool for monitoring water quality and environmental health. The toxicity testing procedures have been designed to assess the effects of both acute [208] and prolonged [209,210,211] exposure of chironomids to waterborne agents. They are basically modifications of the previously described *Daphnia* sp. acute immobilization test. The test uses first instar larvae of *C. riparius* (recommended) or *C. dilutus* and *C. yoshimatsui* that are randomly selected from a batch culture. The larvae are exposed to a determined concentration range of the test substance for 48 h if acute toxicity is measured, and for 44 days in the chronic test. The measured endpoints for the chronic exposure test include the total number of emerged adults, development rate, and fecundity, while the main endpoint in the acute testing protocol is immobilization (mortality rates), as described in the OECD guidelines.

Unlike the filter-feeding zooplankton, larvae of benthic invertebrates often possess strong mandibles used for biting and breaking food apart. This feeding mechanism can potentially increase their exposure to intracellular toxins from cyanobacteria, which are released upon cells’ breakage. A dietary treatment with an MC-LR-producing strain of *Trichormus variabilis* showed no effect on the survival rate; however, significant alterations in oxidative stress enzyme activity and moderate DNA damage were observed [212]. In a later study that used the same cyanobacterial strain, a 40% increase in the mortality of *C. riparius* larvae occurred when exposed to 10 µg/L of MC-LR in an acute toxicity test. The mortality was further increased in the presence of certain environmental stressors, most notably PO_4_^3−^ and Cd^2+^ [213]. Furthermore, chronic exposure of *C. riparius* larvae to the MC-LR producing *T. variabilis* reduced larval mass, hemoglobin concentration, and caused DNA damage in somatic cells. The toxicity of *Plankthothrix agardhii* and *Dolichospermum lemmermannii* extracts containing MC and ANTX-a, as well as the purified forms of MC-LR and ANTX-a, were evaluated using *Chironomus* larvae in 48- and 96-h bioassays [214]. The results showed that the highest concentration of MC-LR (3.32 mg/L) reduced the survival of larvae to 61% after 96 h, while ANTX-a in the same concentration reduced it to 83%. The crude cyanobacterial extracts were found to be significantly more toxic to the exposed organisms, even though MC and ANTX-a concentrations were approximately 10 times lower.

Despite being underrepresented in the literature (Table 2), the use of *Chironomus riparius* as a model organism in cyanobacterial toxicity testing offers several advantages, including its high sensitivity to various toxic compounds, ease of maintenance in laboratory conditions, and the establishment of both acute and chronic testing procedures for this species. This species belongs to the most abundant and widely distributed insect group in freshwater [215] and, as a benthic species, it allows for the functional assessment of the threat posed by sediments polluted by toxic compounds. However, as there are many examples of successful rearing of dipteran larvae using monocultures of cyanobacteria, there is potentially a point to be made for decreased sensitivity of these organisms to cyanobacterial toxicity. Furthermore, differences in sensitivity may exist among different populations of the organism, and its responses may not be representative of other aquatic organisms.

**Table 2 biology-12-00711-t002:** Summary of publications on the use of the most common invertebrate biotests in cyanobacterial toxicity testing.

Model Organism	Observed Parameter	Sample Type	Possible Causative Agent	Exposure Duration	Effective Concentration of the Agent	Reference
*Artemia salina*	Survivorship	Crude extracts— *Microcystis* PCC-7813	MC-LR	18 h	~1 mg/g dw	[156]
		Bloom samples	/	24 h	0.5–5 mg/mL dw	[149]
		Purified toxin	MC-RR		LC50 = 5 µg/mL	
		Bloom samples	MC-LR, MC-RR, NOD	22–24 h	LC50 = 3–17 mg/L	[157]
		Filtered cultures	Anatoxin-a		LC50 = 2–14 mg/L	
	Atypical movement	Fractionated extracts	Anatoxin-a		EC50 = 1–13 µg/L dw	
	Survivorship	Bloom samples	MC-LR	24 h	LC50 = 0.47–2.44 mg dw/L	[132]
		Bloom samples	CYN	48 h	LC50 = 2.8–3.4 ug Chl a/mL	[167]
		*C. raciborskii* extracts	CYN	48 h	LC50 (24 h) = 3.31–5.44 mg/mL dw LC50 (48 h) = 1.68–2.42 mg/mL dw	[164]
		Purified toxin	CYN		LC50 (24 h) = 4.48 µg/mL LC50 (48 h) = 2.86 µg/mL	
		Purified toxin	MC-LR		LC50 (24 h) = 4.58 µg/mL LC50 (48 h) = 2.8 µg/mL	
		Purified toxin	MC-LR	18 h	LC50 (24 h) = 3.75 µg/mL	[169]
		Crude extract	/	24 h	LC50 = 0.7–7.9 mg/mL dw	[216]
		Toxin fraction (concentrated peptides)	/		LC50 = 6.8–12.9 mg/mL dw	
		Purified toxin	MC-LR	18 h	LC50 = 6.8 µg/mL	[170]
		Crude extracts of *M. aeruginosa*	MC		LC50 = 0.8–33.58 mg/mL dw	
		Crude extracts	/	48 h	700–6950 µg/mL dw	[145]
		Crude extracts of *M. aeruginosa* PCC7806	MC-LR		81 ± 3 µg/mL dw	
		Crude extracts	MC-LR	24 h	EC50 = 6.8 ± 2 mg/mL dw	[207]
				48 h	EC50 = 6.8 ± 2 mg/mL dw	
		Toxin fraction (concentrated peptides)	MC-LR	24 h	EC50 = 3.5 ± 1.4 mg/mL dw	
				48 h	EC50 = 2.2 ± 0.7 mg/mL dw	
*Daphnia pulex*	Survivorship	Purified toxin	MC-LR	24 h	LC50 (24 h) > 3.32 mg/L	[192]
		Purified toxin	ANTX-a		LC50 (24 h) > 1.66 mg/L	
*Daphnia magna*		*M. aeruginosa* 7820 cells—ingestion	MC-LR	4 days	3.5 × 10^7^ cells/mL	[184]
*Daphnia pulex*	Survivorship, growth, and reproduction	*M. aeruginosa* 7820 cells—ingestion	MC-LR	21 days	1 × 10^4^–4 × 10^4^ cells/mL	[183]
*Daphnia longispina*						
*Daphnia galeata*	Survivorship	*M. aeruginosa* PCC7806 cells—ingestion	MC-LR	5 days	/	[187]
*Daphnia pulicaria*	Molting disruption	Purified toxin	Microviridin J	4 days	6.75–12 mg/L	[217]
*Ceriodaphnia dubia*	Survivorship	*C. raciborskii* T2	CYN	7 days	197.75 × 10^3^–302.56 × 10^3^ filaments/mL	[118]
		*C. raciborskii* T3			0.218 × 10^3^–5.101 × 10^3^ filaments/mL	
*Ceriodaphnia cornuta*	Population growth	Crude extracts—*Dolichospermum planctonicum*	MC	20 days	0.1180–0.3760 µg/L dw	[203]
*Daphnia similis*	Survivorship	Bloom samples	MC	24 h	LC50 = 186.61 mg/L	[137]
*Ceriodaphnia silvestrii*					LC50 = 155.11 mg/L	
*Daphnia pulex*		Crude extract	/	24 h	LC50 = 0.5–9.2 mg/mL dw	[218]
		Toxin fraction (concentrated peptides)	/		LC50 = 2.01–6.06 mg/mL dw	
*Daphnia magna*		*Cylindrospermopsis raciborskii* cells—ingestion	CYN	72 h	1.8– 5 × 10^5^ cells/mL	[196]
*Daphnia magna*		Crude extract	MC-LR	48 h	EC50 = 6.4 ± 2.3 mg/mL dw	[207]
		Toxin fraction (concentrated peptides)			EC50 = 5.5 ± 0.7 mg/mL dw	
*Daphnia pulex*		Crude extract		24 h	EC50 = 1.1 ± 1.2 mg/mL dw	
		Toxin fraction (concentrated peptides)			EC50 = 1.1 ± 0.4 mg/mL dw	
*Ceriodaphnia dubia*		Crude extract		48 h	EC50 = 6.6 ± 2 mg/mL dw	
		Toxin fraction (concentrated peptides)			EC50 = 6.1 ± 0.4 mg/mL dw	
*Daphnia magna*		Crude extracts	/	48 h	26–75 µg/mL dw	[145]
		Crude extracts of *M. aeruginosa* PCC7806	MC-LR		8 µg/mL dw	
*Daphnia magna*	Survivorship, growth, maturation, time to first reproduction, and fecundity	Purified toxin	MC-LR	2 months	5–50 µg/L	[219]
		MC in crude extracts	MC		5–50 µg/L dw	
*Chironomus riparius*	Survivorship	*Trichormus variabilis* cells—ingestion	MC-LR	12 days	/	[213]
	Oxidative stress induction, DNA damage				5–10 mg of biomass fed every 48 h	
	Survivorship, larval mass reduction, hemoglobin concentration, DNA damage	Purified toxin	MC-LR	48 h	0.01 mg/L	[214]
	Survivorship	Crude extracts—*Plankthothrix agardhii*	MC-LR	96 h	0.42–0.91 mg MC-LR/L dw	[215]
		Purified toxin	MC-LR		1.66–3.32 mg/L	
		Crude extracts—*Dolichospermum lemmermannii*	ANTX-a	96 h	0.12–0.35 mg/L dw	
		Purified toxin	ANTX-a		3.32 mg ANTX-a/L	
*Thamnocephalus platyurus*	Survivorship	Crude extracts—*Microcystis aeruginosa*	/	24 h	0.5–5 mg/mL dw	[205]
		Crude extracts—*Anabaeanaflos-aquae*	/		0.3–5 mg/mL dw	
		Crude extracts—*Cylindrospemopsis raciborskii*	/		1–5 mg/mL dw	
		Crude extracts—*Aphanizomenon flos-aquae*	/		3–5 mg/mL dw	
		Purified toxin	MC-LR	24 h	LC50 = 1.8 mg/L	[138]
		Crude extracts	MC-LR	24 h	LC50 = 0.11 ± 0.3 mg/mL dw	[207]
		Toxin fraction (concentrated peptides)	MC-LR		LC50 = 0.31 ± 0.05 mg/mL dw	

## 4. In Vitro Bioassays

The first in vitro study of cyanobacterial toxicity was published in 1981 [63], guided by the idea that liver deformations previously reported in poisoned animals would be reflected in the effects observed in the isolated hepatic cells. The authors exposed a primary culture of freshly isolated rat hepatocytes to a purified toxin isolated from a *Microcystis aeruginosa* bloom and observed the changes in the affected cells using scanning electron microscopy and phase-contrast microscopy. They found that incubation with the toxin caused cells to become deformed and that the severity of this effect was dependent on the applied dose and duration of exposure. Since this early publication, numerous in vitro studies of cyanobacterial toxicity have been published, and these studies have helped to shed light on the mechanisms of cyanobacterial toxicity and the factors that contribute to this toxicity (Table 3). These in vitro models offer a versatile and flexible approach in cyanobacterial toxicity testing, and they align with the growing trend of minimizing the use of animals in toxicity testing, especially since the establishment of immortalized cell lines, which can renew themselves in artificial cultures indefinitely [220]. Primary cell cultures often maintain enzyme activity to a higher degree than immortalized cell lines, which allows for various investigations to be performed, including the determination of metabolic profiles and examination of inhibition and induction effects. Additionally, a single preparation of primary cells can be used to test a large number of samples, which can be more cost-effective than using whole animals [221]. Established cell lines, on the other hand, have the advantage of being more stable and easier to maintain over time compared with primary cells. This makes them more convenient to use in long-term studies or high-throughput screens [222]. Additionally, established cell lines are readily available and well-characterized, which may allow for more consistent and reliable results compared with primary cells [220]. Due to these advantages, vertebrate cell cultures are an increasingly popular replacement for animal testing in toxicity studies.

However, it should be noted that both primary and established cell lines have limitations. Established cell lines, while stable and well-characterized, may not be fully representative of the target tissue or organ, particularly if the cell line was derived from a different tissue or species [223]. Furthermore, there is evidence of altered transport properties in certain cells due to the passaging process, which leads to decreased carrier permeability for specific compounds, thereby lowering their effectiveness [224].

Primary cell lines, on the other hand, are more difficult to maintain, prone to genetic drift, and have a limited lifespan [225]. Additionally, primary cells can be difficult to obtain, and the process of isolating them can potentially affect their response to toxins. Overall, the use of cell lines in cyanobacterial toxicity testing should be carefully considered, taking into account the specific goals of the study, the availability of appropriate cell lines, and the limitations of each system. In eukaryotic cells, MCs enter via transmembrane multispecific organic anion transporters, which are expressed in the liver, kidney, gastrointestinal tract, and brain [226,227], and once they enter the organism, they accumulate in the liver. Considering that MCs and CYNs, two of the most toxicologically relevant cyanotoxins, induce hepatic damage, many studies concerning cyanobacterial cytotoxicity were selected liver cell lines such as the human hepatocellular carcinoma HepG2. A higher toxicity of CYN than MC-LR in HepG2 cells was reported, with EC50 values (24 h) of ~4 and 90 μg/mL, respectively [228]. The obtained EC50 values in the case of CYN after 24 and 48 h were comparable with those obtained earlier [229,230], while in the case of MC-LR, cytotoxic effects were not found in HepG2 and other hepatic cell lines at concentrations up to 100 μg/mL (48–96 h) [231]. Such discrepancies in EC50 values could be due to experimental conditions (passage of cells, medium used, etc.) [227]. In terms of the cytotoxicity of cyanobacterial extracts to the HepG2 cell line, IC50 values in the range from 49 to 396 μg/mL were recorded [232]. Similar results (35–702 μg/mL) were obtained in later work [145]. In this study, the absence of MCs confirmed via ELISA test indicated that the observed cytotoxicity of extracts could not be attributed to hepatotoxins, while comparison of IC values in the HepG2 cell line with those obtained in three model organisms (*D. magna*, *A. salina*, and zebrafish embryos) showed that the HepG2 cell line is a particularly suitable model for cyanobacterial toxicity assessment. In regard to toxic cyanobacterial blooms, HepG2 cells were used in conjunction with three other cell lines to demonstrate the value of cytotoxicity assays in predicting potential biological hazards in contaminated waters [233]. This is due to the fact that chemical analysis alone can only detect the presence of cyanotoxins. Non-hepatic cell lines, including cancer cell lines, have also been utilized for evaluating cyanobacterial cytotoxicity (refer to Table 3). Among these, HeLa cells (human cervical epithelial adenocarcinoma cells) are most commonly used for conducting cytotoxicity and antitumor activity tests [234]. HeLa cells were found to be the most sensitive to extracts of hepato- and saxitoxins-producing *Fischerella major* (IC50 value of 26.8 µg/mL), compared with the SK-Hep-1 liver cell line and the non-tumor cells FL (human epithelial-like amniotic normal cells), which also showed sensitivity [234]. These findings suggest that the extracts exhibit cytotoxicity rather than a selective anti-tumor effect, since an ideal anticancer treatment should only act against tumor cells, without harming healthy cells. The extracts of the MC- and saxitoxin-producing *Nostoc microscopicum* strain have also exhibited significant cytotoxic activity against HeLa, FL, and A549 (derived from lung carcinoma) cell lines [235]. Similarly, over 100 crude cyanobacterial extracts were tested in six different cell models, and it was found that the sensitivity of HeLa cells was similar to that of HepG2 cells and primary cultures of human hepatocytes (HH) [236]. Although cytotoxicity against HepG2 cells was found in all tested cyanobacterial extracts in one study, MCs were not confirmed via ELISA test, indicating that cytotoxicity cannot be due to hepatotoxins [145]. Interesting findings have also emerged from studies involving cancer cell lines. For instance, almost all tested strains (28) of cyanobacteria induced cytotoxicity in at least one of eight human cancer cell lines [237]. Similarly, strong activity against four different human cancer cell lines (Caco-2, MCF-7, HepG2, and PC3) with the lowest IC50 values of 58 µg/mL, 15 µg/mL, 49 µg/mL, and 44 µg/mL, respectively, was observed when testing 12 species of cyanobacteria [232]. Certain extracts exhibited antiproliferative activities on various cancer cell lines, with the lowest IC50 of 113 µg/mL, in some cases surpassing the anticancer drug fluorouracil in potency [238]. Furthermore, it was demonstrated that in the case of strongly cytotoxic extracts, cytotoxicity is, rather, caused by compounds that are specific for the strain [236]. A CYN-containing extract of *Aphanizomenon ovalisporum* showed a more toxic effect on HepG2 cells than pure toxin CYN (with a 2–2.5-fold lower EC50 for 24 and 48 h), while the extract of the non-CYN-producing *Cylindrospermopsis raciborskii* strain did not show any cytotoxicity [228].

As cyanobacteria are known producers of neurotoxic compounds, some cell lines have been specifically used to estimate neurotoxicity. Human neuroblastoma cells SH-SY5Y have been used in several studies to test the cytotoxicity of BMAA (β-N-methylamino-L-alanine), a non-proteinogenic and toxic amino acid that may harm the nervous system and provoke neurodegenerative diseases such as Alzheimer’s and amyotrophic lateral sclerosis. Using this cell line, different effects were found, such as increased ROS and protein oxidation, upregulation of lysosomal enzymes and apoptosis, misincorporation of L-BMAA protein aggregation, etc., while in HepG2 and Caco-2 cells, BMAA did not affect the common proteinogenic amino acid metabolic pathways [239]. Furthermore, the mouse neuroblastoma cell line Neuro-2A has been developed and used as a screening assay for the determination of saxitoxin toxicity in freshwater cyanobacteria, with an advantage over chromatographic methods that cannot quantify unidentified toxins [240]. A study to test the effects of anatoxin-a at low doses (0.1, 10 µM) was conducted on Neuro-2A cells [241]. It was observed that cell viability decreased to approximately 50% after only 72 h. In murine macrophage-like RAW246.7 cells, a weaker effect was observed, while murine microglial BV-2 cells from the central nervous system were the least sensitive. It was shown that a mixture of CYN, MC-LR, and ANTX-a were 3–15 times more potent at inducing apoptosis and inflammation in immune and brain cells than individual toxins. Furthermore, *Fischerella major* extracts containing MCs and saxitoxins exhibited strong cytotoxic effects on SK-Hep-1 and HeLa cell lines, with IC50 values of 32 and 27 μg/mL, respectively [233]. In another study, it was discovered that the extract of MC- and saxitoxin-producing *Nostoc microscopicum* exhibited strong cytotoxicity to cell lines HeLa, A549, and FL [235].

In contrast to the high sensitivity observed in human hepatic cells, some fish hepatic cell lines, such as RTL-W1 (the rainbow trout liver cell line), commonly used in ecotoxicology [242], were not sensitive under routine conditions when tested with cyanobacterial extracts [145,243,244]. Cytotoxicity against human HepG2 cells was detected with all tested cyanobacterial extracts, but only with the MC-LR-producing strain *Microcystis* PCC 7806 and one *Nostoc* strain in RTL-W1 cells [145]. Similarly, all tested cyanobacterial extracts caused either low or no toxicity in the fish hepatoma PLHC-1/wt cell line derived from topminnow [245].

Monolayer cultures of hepatocellular carcinoma (HCC)-derived or immature human hepatic cell lines, such as HepG2 or undifferentiated HepaRG, have been found to be less sensitive to MC-LR and CYN than primary liver cells. However, three-dimensional (3D) in vitro cultures have been shown to preserve an in-vivo-like phenotype of cultured primary hepatocytes, restore liver-specific functions in HCC-derived cell lines, and facilitate hepatic differentiation of human pluripotent cells, liver stem cells, and progenitor cells [246]. To evaluate the hepatotoxic potential of MC-LR and CYN, one study used 3D cultures of adult human liver stem cells derived from normal, noncancerous tissue, and a telomerase-immortalized HL1-hT1 cell line [246]. It was observed that these spheroid cultures were sensitive to both cyanotoxins (<0.1 µM), which was comparable to the behavior of cultures of primary hepatocytes. It was suggested to use these hepatospheroids for assessing the hepatotoxic potential and monitoring of toxic cyanobacterial samples. In contrast, in monolayer cultures of these HL1-hT1 cells, MC-LR did not induce cytotoxic effects, while CYN inhibited cell growth and viability (48 h–96 h EC50 ≈ 5.5–0.6 µM/L).

**Table 3 biology-12-00711-t003:** Summary of experiments using cell-based assays in cyanobacterial toxicity testing.

Target Organ/System	Cell Line	Applied Assay	Sample Type	Exposure Duration	Effective Concentration	Observed Effect	Reference
Liver	Rainbow trout liver cell line RTL-W1	Alamar Blue (AB) assay	MC-LR	48 h	EC50 > 2.5 μM	No effect on cell viability at moderate (0.25 μM) and high (2.5 μM) MC-LR concentrations	[62]
		CFDA-AM assay					
		Neutral red (NR) assay					
		Alamar Blue (AB) assay	*Phormidium* extracts (five species) showing symptoms of neuro- and hepatotoxicity in mice	24 h	0.75, 3.75, and 15 mg/mL dw	Little to no effect on cell viability	[244]
		CFDA-AM assay					
		MTT colorimetric assay	Crude cyanobacterial extracts	24 h	4, 100, 400, and 2000 μg/mL dw	Little to no effect on cell viability	[145]
			MC-producing *Microcystis* PCC 7806 strain extract		IC50 = 109.16 μg/mL dw	Decrease in cell viability	
	Human hepatocellular carcinoma cell line HepG2	Alamar Blue (AB) assay	MC-LR	48 h	EC50 > 2.5 μM	No effect on cell viability at moderate (0.25 μM) and high (2.5 μM) MC-LR concentrations	[62]
		CFDA-AM assay					
		Neutral red (NR) assay					
		MTT colorimetric assay	*Microcystis* bloom sample extract	72 h	IC50 (24 h) = 214.8 μg/mL dw IC50 (72 h) = 211.5 μg/mL dw	Decrease in cell viability	[247]
			Purified microginins		25–100 µg/mL	Up to 42% decrease in cell viability	
		Tetrazolium salt reduction—MTS assay	Purified CYN	48 h	EC50 = 3.24 ± 0.73 μg/mL	Cytotoxicity/decrease in cellular viability	[228]
			Purified MC-LR		EC50 = 84.18 ± 4.42 μg/mL		
		Total protein content—TP assay	Purified CYN	48 h	EC50 = 3.47 ± 0.41 μg/mL	Cytotoxicity/decrease in cellular viability	
			Purified MC-LR		EC50 = 88.02 ± 1.34 μg/mL		
		MTT colorimetric assay	Purified CYN	24 h	1–5 µg/mL	Up to ~50% decrease in cell viability	[230]
		Comet assay			0.01–5 µg/mL	DNA damage	
		MTT colorimetric assay	Crude cyanobacterial methanolic extracts	-	IC50 = 9–41 µg/mL dw	Strong cytotoxicity	[248]
		MTT colorimetric assay	Crude cyanobacterial extracts	72 h	EC50 = 49–396 μg/mL dw	Decrease in cell viability	[232]
		MTT colorimetric assay	Crude cyanobacterial methanolic extracts	24 h	35–702 μg/mL dw	Decrease in cell viability	[145]
		MTT colorimetric assay	Crude cyanobacterial extracts	24 h	0.04–2 mg/mL dw	Decrease in cell viability	[249]
		Neutral red (NR) assay	Purified MC-LR	24 h	EC50 = 44 µM	Decrease in cell viability	[250]
			*M. aeruginosa* extract		EC50 = 27 µM dw		
		Tetrazolium salt reduction—MTS assay	Purified CYN	72 h	0.1–5 µg/mL	Concentration-dependent inhibition of cell proliferation	[251]
		Comet assay	Crude extracts of MC-LR containing cyanobacterial blooms	24 h	500 µg/mL dw	A low level of DNA damage	[233]
				48 h	50, 125, and 500 µg/mL dw	Total damage of DNA, total mortality even at low concentrations	
		MTT colorimetric assay	Crude cyanobacterial aquatic extracts	24 h	1:10 (*v*/*v*) dilution	>60% of viable cells in most of the cases	[252]
			Methanolic extracts of two *Jaaginema* strains containing no cyanotoxins	48 h	1:10 and 1:50 (*v*/*v*) dilutions	<10% of viable cells	
	Human epithelial-like liver adenocarcinoma cells SK-Hep-1	MTT colorimetric assay	*Fischerella major* extracts containing microcystins and saxitoxins	72 h	IC50 = 32.4–>100 µg/mL dw	Strong cytotoxic effects	[234]
		Neutral red (NR) assay			10, 50, and 100 μg/mL dw	Toxic effects significantly decreased after 48 and 72 h	
	Human hepatocellular carcinoma cell line HuH-7	MTT colorimetric assay	Crude aquatic cyanobacterial extracts	24 h	1:10 (*v*/*v*) dilutions	>70% of viable cells in almost all cases	[253]
		MTT colorimetric assay	Crude cyanobacterial extracts	48 h	IC50 ≥ 1250 μg/mL dw	Decrease in cell viability	[253]
	Human hepatoma cell line Hep3B	MTT colorimetric assay	Crude cyanobacterial methanolic extracts	24 h	IC50 = 245.93–296.15 μg/mL dw	Decrease in cell viability	[238]
		Lactate dehydrogenase (LDH) assay		-	15, 30, 60, and 120 μg/mL dw	Cytotoxicity up to 40%	
	Human liver stem cells HL1-hT1 (monolayer cultures)	Alamar Blue (AB) assay	MC-LR	96 h	EC50 > 10 μM	No cytotoxic effects	[246]
		CFDA-AM assay					
		Neutral red (NR) assay					
		Alamar Blue (AB) assay	CYN	96 h	EC50 = 0.61 μM	Inhibition of cell growth and viability	
		CFDA-AM assay			EC50 = 2.91 μM		
		Neutral red (NR) assay			EC50 = 0.75 μM		
	Primary fish (rainbow trout) hepatocytes	Alamar Blue (AB) assay	MC-LR	48 h	250 nM	Decrease in cell viability to ~70%	[62]
		CFDA-AM assay				No effect on cell viability	
		Neutral red (NR) assay				Decrease in cell viability to ~30%	
	Primary mouse hepatocytes	Alamar Blue (AB) assay	MC-LR	48 h	25 nM	Decrease in cell viability to ~20%	
		CFDA-AM assay				No effect on cell viability	
		Neutral red (NR) assay				Decrease in cell viability to ~20%	
Kidney	Human kidney cells HEK293	Tetrazolium salt reduction—MTS assay	Purified CYN	48 h	2.5–25 µg/mL	Up to 40% decrease in cell viability	[254]
			Purified MC-LR	48 h	50–200 µg/mL	Up to 20% decrease in cell viability	
	African green monkey kidney cell line—Vero	MTT colorimetric assay	Purified MC-LR	72 h	25–200 µM	Cytotoxicity/decrease in cell viability	[255]
		Lactate dehydrogenase (LDH) assay			100–200 µM		
		Neutral red (NR) assay			12.5–200 µM		
		MTT colorimetric assay	*M. aeruginosa* extract		11–175 µM	Cytotoxicity/decrease in cell viability	
		Lactate dehydrogenase (LDH) assay			22–175 µM		
		Neutral red (NR) assay	Purified MC-LR	24 h	EC50 = 53 µM	Decrease in cell viability	[250]
			*M. aeruginosa* extract		EC50 = 34 µM		
		MTT colorimetric assay	Fractions and subfractions of the cyanobacterial bloom containing MC-LR extract	72 h	LC50 = 40–>200 µg/mL dw	Cytotoxic effects	[233]
		MTT colorimetric assay	Crude cyanobacterial extracts showing prominent cytotoxicity on other cell lines	48 h	1:10 (*v*/*v*) dilution	>60% of viable cells	[249]
		MTT colorimetric assay	Crude methanolic cyanobacterial extracts	24 h	IC50 = 144.97–353.95 μg/mL dw	Decrease in cell viability	[238]
		Lactate dehydrogenase (LDH) assay		-	15, 30, 60, and 120 μg/mL dw	Cytotoxicity up to ~60%	
		MTT colorimetric assay	Crude cyanobacterial extracts	48 h	IC50 ≥ 625 μg/mL dw	Decrease in cell viability	[253]
Colon	Human colon carcinoma cells Caco-2	MTT colorimetric assay	Crude cyanobacterial extracts	72 h	EC50 = 58–640 µg/mL dw	Decrease in cell viability	[232]
		Total protein content—TP assay	Purified CYN	48 h	EC50 (24 h) = 36.5 ± 2.1 µg/mL EC50 (48 h) = 2.0 ± 0.5 µg/mL	Time/concentration dependent reduction in protein content	[256]
		Neutral red (NR) assay			EC50 (24 h) = 19.0 ± 1.3 µg/mL EC50 (48 h) = 10.0 ± 1.7 µg/mL	Up to 45% decrease in cell viability	
		Tetrazolium salt reduction—MTS assay			EC50 (24 h) = 2.5 ± 0.4 µg/mL EC50 (48 h) = 0.6 ± 0.2 µg/mL	Decrease in cell viability	
		Tetrazolium salt reduction—MTS assay	Purified CYN	72 h	0.1–5 µg/mL	Concentration-dependent inhibition of cell proliferation	[251]
		MTT colorimetric assay	Purified MC-LR	72 h	50 µM	Up to 90% decrease in cell viability	[76]
		Dichloro-dihydro-fluorescein diacetate (DCFH-DA) assay		120 min		Significant increase in H_2_O_2_ levels at 30 min before returning to normal at 120 min	
		MTT colorimetric assay	Purified MC-LR	48 h	10 µg/mL	Up to 40% decrease in cell viability	[257]
		Dichloro-dihydro-fluorescein diacetate (DCFH-DA) assay		5 h	0.2–10 µg/mL	Intracellular ROS formation	
		Comet assay		16 h	0.2–10 µg/mL	DNA damage	
		MTT colorimetric assay	Crude aquatic cyanobacterial extracts	24 h	1:10 (*v*/*v*) dilution	>60% of viable cells	[252]
			Methanolic extracts of two *Jaaginema* strains containing no cyanotoxins	48 h	1:10 and 1:50 (*v*/*v*) dilutions	<5% of viable cells	
	Human colorectal carcinoma cell line HCT-116	MTT colorimetric assay	Crude methanolic cyanobacterial extracts	-	IC50 = 8–27 µg/mL dw	Cytotoxicity	[248]
	Human colorectal adenocarcinoma cell line HT-29	MTT colorimetric assay	Crude methanolic cyanobacterial extracts	24 h	IC50 = 180.82–386.73 μg/mL dw	Decrease in cell viability	[238]
		Lactate dehydrogenase (LDH) assay		-	15, 30, 60, and 120 μg/mL dw	Cytotoxicity up to ~80%	
Lungs (respiratory system)	Human fetal lung cell line MRC-5	Colorimetric sulforhodamine B (SRB) assay	Water samples from blooming lakes	48 h	Raw sample diluted to 10%	Up to ~30% decrease in cell viability	[258]
	Human lung adenocarcinoma cell line A549	MTT colorimetric assay	Microcystin- and saxitoxin-producing *Nostoc microscopicum* (acetic acid extract)	24 h	IC50 = 173 μg/mL dw	Prominent cytotoxic activity	[235]
		MTT colorimetric assay	Crude methanolic cyanobacterial extracts	24 h	IC50 = 284.20–407.95 μg/mL dw	Decrease in cell viability	[238]
		Lactate dehydrogenase (LDH) assay		-	15, 30, 60, and 120 μg/mL dw	Low cytotoxicity at all concentrations (~10–20%)	
Endothelium	Human umbilical vein endothelial cell line HUVEC	Total protein content—TP assay	Purified CYN	48 h	EC50 (24 h) = 8.5 ± 1.2 µg/mL EC50 (48 h) = 1.5 ± 0.6 µg/mL	Time/concentration-dependent reduction in protein content	[259]
		Neutral red (NR) assay			EC50 (24 h) = 1.5 ± 0.9 µg/mL EC50 (48 h) = 0.8 ± 0.5 µg/mL	Decrease in cell viability	
		Tetrazolium salt reduction—MTS assay			EC50 (24 h) = 15.5 ± 2.1 µg/mL EC50 (48 h) = 1.5 ± 0.3 µg/mL		
Blood	Human peripheral blood mononuclear cells (PBMCs)	MTT colorimetric assay	Crude cyanobacterial extracts	72 h	EC50 = 28–991 µg/mL dw	Decrease in cell viability	[232]
	Human peripheral blood lymphocytes (HPBL)	Differential staining (acridine orange and ethidium bromide)	Purified MC-LR	24 h	0.1–10 µg/mL	No effect on cell viability	[260]
		Comet assay				DNA damage	
		Comet assay	Purified MC-LR	24 h	1–25 µg/mL	DNA damage	[261]
	Human leukemia cell line HL-60	Lactate dehydrogenase (LDH) assay	Crude methanolic cyanobacterial extracts	3 h	20, 100, and 200 μg/mL dw	Cytotoxicity up to 100%	[262]
	Human promonocytic cells U-937	MTT colorimetric assay	Fractions and subfractions of the cyanobacterial bloom containing MC-LR extract	72 h	LC50 = 17–>200 µg/mL dw	Cytotoxic effects	[233]
	Mouse monocytic cells J774				LC50 = 75–>200 µg/mL dw		
Nervous system	Mouse neuroblastoma—Neuro-2a	Colorimetric sulforhodamine B (SRB) assay	Water samples from blooming lakes	48 h	Raw sample diluted to 10%	Up to ~20% decrease in cell viability	[258]
	Rat glioma cell line C6	MTT colorimetric assay	Crude methanolic extracts	24 h	IC50 = 112.69–164.90 μg/mL dw	Decrease in cell viability	[238]
		Lactate dehydrogenase (LDH) assay		-	15, 30, 60, and 120 μg/mL dw	Cytotoxicity up to ~80%	
Reproductive system	Human breast cancer cell line MCF-7	MTT colorimetric assay	Crude cyanobacterial extracts	72 h	EC50 = 15–361 µg/mL dw	Decrease in cell viability	[232]
		MTT colorimetric assay	Crude methanolic cyanobacterial extracts	-	IC50 = 11–38 µg/mL dw	Decrease in cell viability	[248]
		MTT colorimetric assay	Crude methanolic cyanobacterial extracts	24 h	IC50 = 133.16–189.45 μg/mL dw	Decrease in cell viability	[238]
		Lactate dehydrogenase (LDH) assay		-	15, 30, 60, and 120 μg/mL dw	Cytotoxicity up to ~40%	
		Colorimetric sulforhodamine B (SRB) assay	*Geitlerinema* sp. CNP 1019 strain extract	48 h	GI50 = 25.7 μg/mL dw	Cytotoxicity	[263]
	Human prostate cancer cell line PC3	MTT colorimetric assay	Crude cyanobacterial extracts	72 h	EC50 = 44–339 µg/mL dw	Decrease in cell viability	[232]
	Human cervical adenocarcinoma cell line HeLa	MTT colorimetric assay	Microcystins and saxitoxin-producing *Nostoc microscopicum* (acetic acid extract)	24 h	IC50 = 270 μg/mL dw	Prominent cytotoxic activity	[235]
		MTT colorimetric assay	Fractions and subfractions of the cyanobacterial bloom containing MC-LR extract	72 h	LC50 = 109.5–>200 µg/mL dw	Cytotoxic effects	[233]
		MTT colorimetric assay	*Fischerella major* extracts containing microcystins and saxitoxins	72 h	IC50 = 27–59 µg/mL dw	Strong cytotoxic effects	[234]
		Neutral red (NR) assay			IC50 = 34–95 µg/mL dw	Toxic effects	
		MTT colorimetric assay	Crude methanolic cyanobacterial extracts	24 h	IC50 = 151.36–209.43 μg/mL dw	Decrease in cell viability	[238]
		Lactate dehydrogenase (LDH) assay		-	15, 30, 60, and 120 μg/mL dw	Low cytotoxicity at all concentrations (~10–20%)	
		MTT colorimetric assay	Crude methanolic cyanobacterial extracts	24 h	IC50 = 0.2–>20 mg/mL dw (determined only for selected strains)	20% of tested extracts exhibited strong cytotoxicity	[236]
	Human normal amniotic cells FL	MTT colorimetric assay	Microcystin- and saxitoxin-producing *Nostoc microscopicum* (acetic acid extract)	24 h	IC50 = 253 μg/mL dw	Prominent cytotoxicity	[235]
		MTT colorimetric assay	*Fischerella major* extracts containing microcystins and saxitoxins	72 h	IC50 = 29–62 μg/mL dw	Strong cytotoxic effects	[234]
		Neutral red (NR) assay			10, 50, and 100 μg/mL dw	Toxic effects were detected after 24 h, while the cells were able to overcome these effects after 48 and 72 h	
		MTT colorimetric assay	*Phormidium* extracts (five species) showing symptoms of neuro- and hepatotoxicity in mice	24 h	15 mg/mL dw	Decrease in cell viability over 50%, up to ~20%	[244]
Others	Human dermal fibroblasts (HDF cells)	Tetrazolium salt reduction—MTS assay	Purified CYN	48 h	IC50 > 5 µg/mL	Concentration-dependent inhibition of cell proliferation	[251]
		Lactate dehydrogenase (LDH) assay		72 h	0.1–5 µg/mL	Cytotoxicity reached only 30% at concentrations above 1 μg/mL	
	Mouse embryonic fibroblast cell line 3T3	Lactate dehydrogenase (LDH) assay	Crude methanolic cyanobacterial extracts	3 h	20, 100, and 200 μg/mL dw	Cytotoxicity up to 100%	[262]
		MTT colorimetric assay	*Phormidium* extracts (five species) showing symptoms of neuro- and hepatotoxicity in mice	24 h	15 mg/mL dw	Decrease in cell viability ~50%	[244]
	Human oral cell line KB	Colorimetric sulforhodamine B (SRB) assay	*Geitlerinema* sp. CNP 1019 strain extract	48 h	GI50 = 60.1 μg/mL dw	Cytotoxicity	[263]
	Human metastatic melanoma cell line A2058	MTT colorimetric assay	*Phormidium* extracts (five species) showing symptoms of neuro- and hepatotoxicity in mice	24 h	15 mg/mL dw	Decrease in cell viability over 50%, up to ~20%	[244]
	Human embryonic rhabdomyosarcoma cell line RD						

## 5. Limitations and Challenges in Cyanobacterial Toxicity Testing

Given the complexities of cyanobacterial toxicity, developing alternatives to mammalian assays with high predictability of in vivo effects is a difficult task. Bioassays play a crucial role in the investigation of substances produced by cyanobacteria, which may have unknown or insufficiently characterized effects. It is clear that the field of cyanobacterial toxicity testing faces several challenges, including ethical limitations in using various animal models, the varying sensitivity levels of different model organisms, the difficulty in extrapolating data to humans, the complexity of samples where identifying the specific compound responsible for toxicity is often challenging, and inability to accurately predict long-term effects of low-level exposure. These challenges require innovative approaches to ensure accurate and ethical testing while providing valuable insights into the potential risks of cyanobacterial toxins. The many bioassays that have been developed, from molecular to whole-organism levels, offer different complexities, and each approach has its own strengths and limitations [264,265,266]. However, a single test is often insufficient, and a combination of different testing methods may be necessary to fully assess the potential risks of exposure to various toxic cyanobacterial metabolites. Animal models continue to provide benefits because they share genetic and physiological similarities with humans, however, unforeseeable factors in animal organisms and the high cost of breeding and housing animals for research purposes are crucial aspects to consider [267,268]. On the other hand, human cell cultures can be cultivated as organotypic cultures to permit easier extrapolation of in vitro results to humans. However, obtaining and treating basic human cells in a safe and ethical manner remains a challenge. Furthermore, in vitro testing may not accurately depict the complexity of whole organisms and the interactions of diverse cell types and organs [269]. Despite their rapid responses, it is essential to validate bioassays’ performances, particularly if they are to be used in complex samples such as raw or treated drinking water or blooms. In vitro bioassays can provide insight into the biochemical processes underlying toxicity, while in vivo studies, despite ethical and technical concerns, are still necessary for risk assessment and guideline value derivation [270,271].

## 6. Tracking the Evolution of Bioassays for Cyanotoxin Testing

In this section, an overview of the historical and current research landscape in the field of cyanobacterial toxicity testing is provided. A comprehensive search of three major scientific databases, i.e., Scopus, PubMed, and Embase was conducted, focusing on publications that utilize some the most commonly applied animal models in cyanobacterial toxicity testing (Figure 1). The obtained references were categorized according to the year they were published, providing an insight into the historical development of bioassays for cyanotoxin testing and the frequency of their use from 1969 until today. Furthermore, a distribution of publications across different toxin classes was included, allowing for a visual representation of the current state of research in the field of cyanotoxin bioassay testing, and highlighting areas where further research would be needed. The main cyanotoxin classes included in the analysis were microcystins (MCs), cylindrospermopsins (CYN), all anatoxins, and guanitoxin grouped together and presented as ANTX and saxitoxins (STX).

After combining the found publications from all three databases into a single document, duplicates and secondary source publications were removed, resulting in a total of 1067 original research articles used in our analysis. Among these, *Artemia salina* were used in 176 publications, with 38 studies focusing on microcystin (MC) testing, 10 on cylindrospermopsin (CYN), 6 on saxitoxin (STX), 3 on anatoxins (ANTX), and 119 on other toxic metabolites or uncharacterized samples such as crude extracts. *Daphnia* were the most commonly used model, with 446 publications utilizing various species of daphnids. Of these, 198 studies focused on MC testing, 41 on CYN, 14 on STX, 9 on ANTX, and 184 on extracts or other toxic metabolites. The mouse bioassay was used in 322 publications, with 181 focusing on MC testing, 33 on CYN, 16 on STX, 18 on ANTX, and 74 on other toxins or uncharacterized samples. Finally, zebrafish were used in 123 publications, with 71 studies focusing on MC testing, 11 on CYN, 12 on STX, 3 on ANTX, and 26 on other toxic metabolites or uncharacterized samples.

The analysis of the number of publications over time revealed a gradual decline in the use of mice for cyanobacterial toxicity testing, particularly after the late 1990s and early 2000s. Simultaneously, there has been a significant increase in the utilization of alternative vertebrate and invertebrate models included in the analysis. This trend corresponds with some of the early articles and advisory statements that paved the way for the development and validation of alternative methods for toxicity testing of water samples, such as in vitro assays and bioassays using other non-mammalian model organisms [272]. The utilization of the *Daphnia* bioassay has significantly increased since the adoption of toxicity testing guidelines for these species in 2004 [178] and 2012 [179], with over 20 studies being published annually since 2017 and continuing to this day. The most-studied cyanotoxins in all included assays were microcystins and cylindrospermopsin, which was expected given their toxicity and wide distribution.

## 7. Conclusions

The research field of cyanobacterial toxicology has greatly benefited from the use of bioassays, as they have provided a more complete understanding of the diverse mechanisms of cyanotoxin action. Most studies concerned with the effects of cyanotoxins are conducted by observing changes occurring in a living organism after exposure to toxins in some form, usually a purified single toxin, although more complex crude extracts and bloom biomass are frequently analyzed. Exploring the effects of unknown or not sufficiently characterized substances produced by cyanobacteria is crucial, and bioassays are an important tool for achieving this. From molecular to organism levels, a wide range of bioassays are available. Some of them, such as the *Artemia salina* assay, offer a simple and inexpensive solution for rapid screening of a large number of samples containing potentially toxic compounds, without sacrificing reliability and sensitivity of the test, while others, such as the zebrafish embryo assay, can provide the means for a more in-depth analysis of toxicity on different biological levels. However, any single test is usually insufficient to fully characterize the toxicity of a cyanobacterial bloom; thus, a combination of the appropriate ones should be employed to achieve an accurate estimation. Bioassays can provide quick responses, but they require an understanding of the sensitivity limitations of the bioassays. Validation of the suitability of the chosen method is necessary, particularly when used to investigate critical samples such as raw or treated drinking water and complex samples such as cyanobacterial blooms. In vitro bioassays are useful in developing an understanding of the biochemical processes underlying toxicity, while in vivo studies, despite technical and ethical concerns, continue to play an important role in supporting risk assessment and guideline value derivation. In contrast to in vitro assays, which provide a simplified and isolated view of a toxicant’s impact, in vivo assays offer a more comprehensive representation of the complex interactions that occur in living organisms. Moving forward, there is a need for further research to refine and optimize vertebrate bioassays for cyanotoxicity testing. This includes developing standardized protocols for bioassays that can be used across different laboratories and regions, as well as identifying novel model organisms (including microorganisms and plants) that can provide new insights into the mechanisms of cyanotoxicity and which have fewer ethical concerns. Additionally, new technologies such as in vitro models and computational modeling can complement vertebrate bioassays and help reduce the number of animals used in research.

## Figures and Tables

**Figure 1 biology-12-00711-f001:**
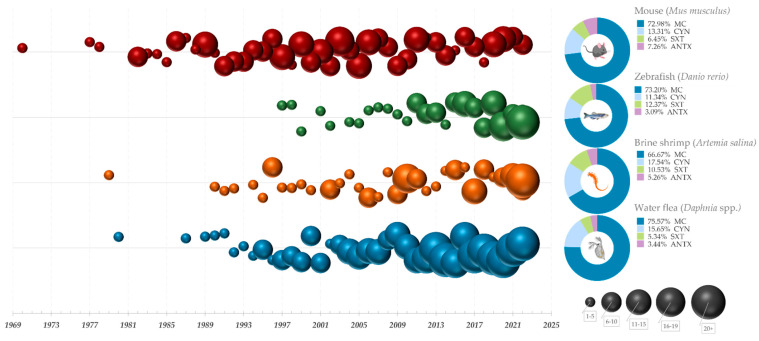
A historical overview of bioassays used in cyanobacterial toxicity testing. Graphs on the left side represent timelines for some of the most widely used bioassays, with the number of publications from each year, starting from 1969, represented by bubbles of different sizes and colors (Mouse–red; Zebrafish–green; Brine shrimp–orange; Water flea–blue). Legend in the lower right corner signifies 5 different bubble categories divided according to the number of publications they represent. The pie charts on the right depict the proportion of publications relating to major cyanotoxin groups.

## Data Availability

Not applicable.

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
