# Peer review of "Biotests in Cyanobacterial Toxicity Assessment—Efficient Enough or Not?"

_biology, 2023, doi:10.3390/biology12050711_

Round 1

Reviewer 1 Report

In this paper, the authors provide a thorough review of various animal bioassays conducted on cyanotoxins and the challenges/needs for a multi-level approach to studying associated toxicity. The authors propose continued development of bioassays such as the in vitro methods as necessary to better improve the detection and characterization of cyanotoxins in animals.

This review paper addresses an important topic which up until recently has been addressed somewhat incompletely. The authors place particular emphasis on in vitro bioassays, which may play a key role in supporting risk assessment. It has long been postulated that bioassays using mammalian cells could be an appropriate substitute for animal bioassays. However, it is beneficial for this review paper to address other advances including -OMICs approaches (e.g., proteomics, see references below) which have been recently employed within routine toxicity risk assessments to better characterize the mechanisms of toxicity from cyanobacteria.

-Shahmohamadloo, R. S., Simmons, D. B., & Sibley, P. K. (2020). Shotgun proteomics analysis reveals sub-lethal effects in Daphnia magna exposed to cell-bound microcystins produced by Microcystis aeruginosa. Comparative Biochemistry and Physiology Part D: Genomics and Proteomics, 33, 100656.

-Shahmohamadloo, R. S., Almirall, X. O., Simmons, D. B., Poirier, D. G., Bhavsar, S. P., & Sibley, P. K. (2022). Fish tissue accumulation and proteomic response to microcystins is species-dependent. Chemosphere, 287, 132028.

Admittedly, the above studies have been conducted on whole organisms but are an important feature in the continued development and refinement of cyanotoxicity bioassays. A major reason for the inclusion of -OMICs approaches within this review paper is because proteomics, for example, describes at the protein level whole-organism responses to cyanotoxicity, which help to provide a mechanism explaining injury from sublethal exposure. Furthermore, -OMICs approaches are becoming more cost-effective, and can describe a plethora of stress responses directly and indirectly occurring due to cyanotoxicity. For instance, in the section on Daphnia it may benefit from a multigenerational, proteomics perspective showing that Daphnia offspring are impaired from their mothers who were exposed to sublethal concentrations of Microcystis (see reference above). This type of information better helps laboratory toxicity assays make reliable predictions of population-level consequences occurring in the field. Ultimately, one goal of risk assessment from laboratory bioassays is to make reliable field predictions. There are certainly many other -OMICs papers in the literature that can be included but these two are provided here as a starting point. 

One minor suggestion is made for L#159-161. This statement is not necessarily true. Non-targeted analysis can detect peaks corresponding with unknown toxic agents, albeit not characterized by still detected.

Author Response

Dear editor,

Thank you for giving us the opportunity to submit a revised manuscript titled „Biotests in cyanobacterial toxicity assessment- efficient enough or not?“ for publication in the journal Biology. We appreciate the time and effort that you and the reviewers dedicated to providing feedback on our manuscript and are grateful for the insightful comments. In the revised version we have adressed all of the comments and suggestions made by reviewers. We believe that the contents and the clarity of our paper are much improved in the revised version. All changes are highlighted within the manuscript. Please see the responses to the reviewers’ comments below.

Reviewer #1:

  1. This review paper addresses an important topic which up until recently has been addressed somewhat incompletely. The authors place particular emphasis on in vitro bioassays, which may play a key role in supporting risk assessment. It has long been postulated that bioassays using mammalian cells could be an appropriate substitute for animal bioassays. However, it is beneficial for this review paper to address other advances including -OMICs approaches (e.g., proteomics, see references below) which have been recently employed within routine toxicity risk assessments to better characterize the mechanisms of toxicity from cyanobacteria.

-Shahmohamadloo, R. S., Simmons, D. B., & Sibley, P. K. (2020). Shotgun proteomics analysis reveals sub-lethal effects in Daphnia magna exposed to cell-bound microcystins produced by Microcystis aeruginosa. Comparative Biochemistry and Physiology Part D: Genomics and Proteomics, 33, 100656.

-Shahmohamadloo, R. S., Almirall, X. O., Simmons, D. B., Poirier, D. G., Bhavsar, S. P., & Sibley, P. K. (2022). Fish tissue accumulation and proteomic response to microcystins is species-dependent. Chemosphere, 287, 132028.

Admittedly, the above studies have been conducted on whole organisms but are an important feature in the continued development and refinement of cyanotoxicity bioassays. A major reason for the inclusion of -OMICs approaches within this review paper is because proteomics, for example, describes at the protein level whole-organism responses to cyanotoxicity, which help to provide a mechanism explaining injury from sublethal exposure. Furthermore, -OMICs approaches are becoming more cost-effective, and can describe a plethora of stress responses directly and indirectly occurring due to cyanotoxicity. For instance, in the section on Daphnia it may benefit from a multigenerational, proteomics perspective showing that Daphnia offspring are impaired from their mothers who were exposed to sublethal concentrations of Microcystis (see reference above). This type of information better helps laboratory toxicity assays make reliable predictions of population-level consequences occurring in the field. Ultimately, one goal of risk assessment from laboratory bioassays is to make reliable field predictions. There are certainly many other -OMICs papers in the literature that can be included but these two are provided here as a starting point.

Response: We appreciate the reviewer's insightful comments regarding the potential benefits of including OMICs approaches in this paper. We agree that such approaches can provide valuable insights into the underlying mechanisms. As suggested, we have expanded the discussion of -OMICs approaches in several sections of the paper, beginning with the Daphnia section in lines 861-879. In addition, we have included references to other OMICs studies in the literature to provide a more comprehensive overview of this rapidly evolving field. We would like to clarify that the section about fish models already contains information about OMICs approaches, but now has additional articles added on this topic.

  1. One minor suggestion is made for L#159-161. This statement is not necessarily true. Non-targeted analysis can detect peaks corresponding with unknown toxic agents, albeit not characterized by still detected.

Response:  We fully agree with the reviewer’s suggestion. The statement has been deleted and a new paragraph has been introduced in lines 159-173, where we have expanded on the function of analytical measurement and highlighted its significance in determining and monitoring the presence of cyanotoxins in water bodies.

Reviewer 2 Report

This manuscript provides a summary of biotest data for cyanobacterial toxicity assessment. The manuscript feels like an opinion piece that in vitro is better than vertebrate test. The manuscript should be revised to only present facts and remove any opinions. Also, the role of analytical measurement of cyanotoxins is not stressed enough.  The source and verification of samples used in each study mentioned should be called in to question.  My final major concern over this paper is the lack of endpoint discussion.  In vitro can generate a lot of data, but this data cannot be used to set drinking water levels.  This discussion needs to be included in the manuscript. 

Lines 185-186: Provide a reference for the statement “vertebrate bioassays can be time-consuming and expensive” or delete the statement.  This feels more like opinion than a fact.  Aren’t in vitro experiments also expensive and time consuming?

Lines 187-188: This limitation due to sample matrix is not specific to vertebrate assays, but would apply to all types of studies.  This statement should be deleted or moved to another section to acknowledge the overall problem with sample matrix and not try to link it to vertebrate studies. 

Lines 193-195: If exposure route is known to be important then how can in vitro ever be considered valid?

Lines 197-198: There are several oral dosed mouse studies e.g. https://pubmed.ncbi.nlm.nih.gov/32570788/ https://pubmed.ncbi.nlm.nih.gov/33498948/

Author Response

Dear editor,

Thank you for giving us the opportunity to submit a revised manuscript titled „Biotests in cyanobacterial toxicity assessment- efficient enough or not?“ for publication in the journal Biology. We appreciate the time and effort that you and the reviewers dedicated to providing feedback on our manuscript and are grateful for the insightful comments. In the revised version we have adressed all of the comments and suggestions made by reviewers. We believe that the contents and the clarity of our paper are much improved in the revised version. All changes are highlighted within the manuscript. Please see the responses to the reviewers’ comments below.

Reviewer #2:

  1. The manuscript feels like an opinion piece that in vitro is better than vertebrate test. The manuscript should be revised to only present facts and remove any opinions.

Response: We understand your concern that our manuscript may have come across as presenting opinions rather than objective facts. We have gone through the entire manuscript and corrected any statements that could be construed as stating that in vitro testing is inherently superior to vertebrate testing. We have particularly focused on the abstract section that contains a summary of the findings, as well as the sections where we discuss in vitro and vertebrate assays.

  1. Also, the role of analytical measurement of cyanotoxins is not stressed enough.

Response:  We agree that it is an essential aspect that needs to be highlighted more explicitly. We have now revised the manuscript to stress the importance of analytical measurement in detecting and quantifying cyanotoxins, particularly in environmental samples. In section [173-191], we have elaborated on the role of analytical measurement, highlighting its significance in identifying and monitoring the presence of cyanotoxins in water bodies.

  1. The source and verification of samples used in each study mentioned should be called in to question.

Response: We agree with the point made by the reviewer and have added two sentences about this in the new text about analytical methods ("Besides validation of the analytical robustness itself the analyst should pay attention to receiving representative and non-tampered samples. Ideally, critical samples such as samples of potable water should be analysed by two or more independent methods based on different chemical, biochemical or toxicological principles."). When it comes to assessing all the details in the reviewed publications, it is practically impossible to exhaustively evaluate the source and verification of the samples used. However, only peer-reviewed papers have been used to construct the present review and this gives a reasonable amount of confidence to the work.

  1. My final major concern over this paper is the lack of endpoint discussion. In vitro can generate a lot of data, but this data cannot be used to set drinking water levels. This discussion needs to be included in the manuscript.

Response: We appreciate the reviewer’s suggestion and fully agree with the claim. We have added a new section in the manuscript to discuss this topic in more detail in lines 1128-1153. We have also included a graphical abstract that highlights the strengths and limitations of different approaches, including in vitro, to help readers understand the distinction better.

  1. Lines 185-186: Provide a reference for the statement “vertebrate bioassays can be time-consuming and expensive” or delete the statement. This feels more like opinion than a fact. Aren’t in vitro experiments also expensive and time consuming?

Response:  We thank the reviewer for drawing our attention to this inconsistency. We understand the concern about the lack of supporting evidence and acknowledge that both vertebrate bioassays and in vitro experiments can be costly and time-consuming. In light of this, we have decided to delete this statement from our manuscript to ensure that our work remains objective.

  1. Lines 187-188: This limitation due to sample matrix is not specific to vertebrate assays, but would apply to all types of studies. This statement should be deleted or moved to another section to acknowledge the overall problem with sample matrix and not try to link it to vertebrate studies.

Response: Considering the reviewer`s feedback, we have decided to remove this statement from our manuscript as the problem indeed is not connected solely to vertebrate studies.

  1. Lines 193-195: If exposure route is known to be important then how can in vitro ever be considered valid?

Response: We thank the reviewer for the opportunity to clarify our statement. Our intention was to highlight the variability in uptake rates of cyanotoxins depending on the exposure route, which can impact the amount of toxin accumulated in an organism. We acknowledge that in vitro tests have limitations, but they can still provide valuable mechanistic information about toxins and their effects on target tissues. Furthermore, when used in conjunction with other methods, they can contribute to a more comprehensive understanding of toxicity. We have added additional clarification in the text to emphasize these points.

  1. Lines 197-198: There are several oral dosed mouse studies e.g. https://pubmed.ncbi.nlm.nih.gov/32570788/ https://pubmed.ncbi.nlm.nih.gov/33498948/

Response:  We appreciate the suggestion on lines 197-198 and we have amended our statement to include oral exposure as a possible route of cyanobacterial toxin exposure.

Reviewer 3 Report

Biotests in cyanobacterial toxicity assessment- efficient enough or not? are auspicious as contamination in cyanobacterial toxicity. However, English does not sound very good. You may need to hire a professional English editing service or have your paper proofread by a native English-speaking colleague.

The abstract should be rewritten, not matched.

I have not found any introduction section in the manuscript. Include it.

The hypothesis is not explained clearly.

The author should include figures about the toxicity test.

Include a figure of the main finding of this manuscript.

There are no future directions for the work mentioned.

Write a paragraph about the limitation of toxicity assessment.

The present form of the manuscript cannot be published prior to major revision.

Author Response

Dear editor,

Thank you for giving us the opportunity to submit a revised manuscript titled „Biotests in cyanobacterial toxicity assessment- efficient enough or not?“ for publication in the journal Biology. We appreciate the time and effort that you and the reviewers dedicated to providing feedback on our manuscript and are grateful for the insightful comments. In the revised version we have adressed all of the comments and suggestions made by reviewers. We believe that the contents and the clarity of our paper are much improved in the revised version. All changes are highlighted within the manuscript. Please see the responses to the reviewers’ comments below.

Reviewer #3:

  1. Biotests in cyanobacterial toxicity assessment- efficient enough or not? are auspicious as contamination in cyanobacterial toxicity. However, English does not sound very good. You may need to hire a professional English editing service or have your paper proofread by a native English-speaking colleague.

Response:  We thank the reviewer for the feedback. We appreciate the suggestion regarding the quality of English in our manuscript. We have carefully reviewed and revised the manuscript, and we believe that the language is now clear and understandable.

  1. The abstract should be rewritten, not matched.

Response: We appreciate the feedback and have revised the abstract to better reflect the theme and title of the article, as suggested.

  1. I have not found any introduction section in the manuscript. Include it.

Response:  We appreciate the valuable input and have made the necessary changes to improve the clarity of our work. We would like to clarify that we have renamed the section previously titled "The problem of cyanobacterial toxicity" to simply "Introduction" to make it more explicit.

  1. The hypothesis is not explained clearly.

Response:  We thank the reviewer for the feedback. As a review article, we appreciate that our work does not have a formal hypothesis in the same way that an original research article would. However, we have taken your comments on board and have reworded the goals section of the manuscript (lines 128-135) to outline our main objectives more clearly. We hope that this revised section more effectively conveys the scope and focus of our article.

  1. The author should include figures about the toxicity test.

Response: We understand the importance of providing visual aids to better illustrate our findings and thank the reviewer for the valuable suggestion. However, we also want to avoid overburdening the text with excessive figures. To address this concern, we have created a graphical abstract that encompasses each group of bioassays mentioned in the article. We believe that this approach will provide a clear and concise overview of the toxicity tests and their outcomes.

  1. Include a figure of the main finding of this manuscript.

Response: We thank the reviewer for the suggestion. We have prepared a graphical abstract that presents the main conclusions of the article in a visual and easily accessible format.

  1. There are no future directions for the work mentioned.

Response: We appreciate the feedback provided by the reviewer and have carefully considered their comment regarding the future directions of our work. While we did mention the future directions in the conclusion section (lines 1226-1233), we agree that it would be beneficial to make them more visible and clear to the reader. To address this, we have revised the abstract section to include a clearer statement of the future directions we propose.

  1. Write a paragraph about the limitation of toxicity assessment.

Response: We thank the reviewer for the suggestion. As such, we have added a section in lines 1128-1153 to discuss these limitations.

Round 2

Reviewer 1 Report

The authors have thoughtfully considered my reviewer comments and applied where necessary. The paper reads well and is suitable for publication.

Reviewer 3 Report

Author did all my comments meticulously.